# Putting your money where your self is: Connecting dimensions of closeness and theories of personal identity

Jan K. Woike[1]*, Philip Collard[2], Bruce Hood[2]

1 Max Planck Institute for Human Development, Center for Adaptive Rationality (ARC), Berlin, Germany,
2 University of Bristol, School of Psychological Science, Bristol, United Kingdom

* woike@mpib-berlin.mpg.de

## Abstract

Studying personal identity, the continuity and sameness of persons across lifetimes, is notoriously difficult and competing conceptualizations exist within philosophy and psychology. Personal reidentification, linking persons between points in time is a fundamental step in allocating merit and blame and assigning rights and privileges. Based on Nozick's (1981) closest continuer theory we develop a theoretical framework that explicitly invites a meaningful empirical approach and offers a constructive, integrative solution to current disputes about appropriate experiments. Following Nozick, reidentification involves judging continuers on a metric of continuity and choosing the continuer with the highest acceptable value on this metric. We explore both the metric and its implications for personal identity. Since James (1890), academic theories have variously attributed personal identity to the continuity of memories, psychology, bodies, social networks, and possessions. In our experiments, we measure how participants ($N = 1,525$) weighted the relative contributions of these five dimensions in hypothetical fission accidents, in which a person was split into two continuers. Participants allocated compensation money (Study 1) or adjudicated inheritance claims (Study 2) and reidentified the original person. Most decided based on the continuity of memory, personality, and psychology, with some consideration given to the body and social relations. Importantly, many participants identified the original with both continuers simultaneously, violating the transitivity of identity relations. We discuss the findings and their relevance for philosophy and psychology and place our approach within the current theoretical and empirical landscape.

## Introduction

In September 1560, Arnaud du Tilh was hanged in front of a house belonging to Martin Guerre, whose name and place in life he had stolen for three years, living with Guerre's wife and children and taking ownership of his possessions [1, 2]. When the true Martin Guerre came back from the war and faced his double in a trial, his presence was sufficient evidence for sentencing the impostor to death.

**Data Availability Statement:** The data required to replicate the results of this study for both Study 1 and Study 2 can be accessed in the Harvard Dataverse (https://doi.org/10.7910/DVN/AN1QF3). The dataset for Study 1 is freely accessible in a

version without demographic information and can be requested via the platform with demographic data by researchers at academic institutions upon reasonable request (the full dataset is uploaded to the platform).

**Funding:** This work was supported by the Economic and Social Research Council, UK (grant number ES/K010131/1, BH, https://esrc.ukri.org) and the John Templeton Foundation (grant number S-000608, BH, PC, https://www.templeton.org/). The funders had no role in study design, data collection and analysis, decision to publish, or preparation of the manuscript.

**Competing interests:** The authors have declared that no competing interests exist.

Most accept the fact that two people cannot have a legitimate claim to be one and the same person. Monozygotic twins may share all relevant features and be close to qualitatively identical at birth—they share all qualitative features like two black billiard balls fresh out of production (see [3], p. 5). But just as the two billiard balls cannot take up the same position in time and space or become one and the same object, these twins will never be numerically identical [4, 5]; they will go through distinct experiences and remain two different people, with different origins and life histories.

This is the standard way personal identity is treated in philosophy, where it does not consist of a list of qualitative properties but rather refers to the numerical sameness of persons across time or observational perspectives. I may consider myself to be the numerically same person as I was five years ago, even if I can detect qualitative differences in myself then and now. Likewise, I don't expect a billiard ball that I paint in a different color to become a numerically different ball (see [6], p. 201). Parfit [7] and many others holds that identity consists in these very continuities and is not a further fact added to them. The identity of objects and the identity of persons thus refers to the same type of sameness or continuity. This sense of identity is far from trivial, as the struggle for resolution in the Martin Guerre example shows.

Personal identity is not only of legal relevance for determining ownership but also of intellectual interest in probing and defining the concepts of persons and the individual. Personal identity—in terms of the continuity of persons—is indeed considered essential for any justification of rewards and punishments [8, 9], future self-concern, moral responsibility, pride, regret, or nostalgia [10]. We probably could not have a sensible notion of "person" without some form of personal continuity. For Locke (see [9], p. 223–224), "person" was a "forensic term" that connects actions and their merits and features centrally not only in law, but also in feelings of accomplishment, regret, praise, and blame [11]. The tracking of persons is therefore at the heart of institutional systems such as criminal law, property, credit, and insurance, determining both responsibility and entitlement [12]. Personal identity serves as an organizing principle that makes temporal perspectives possible; it is necessary for expectations, planning, and episodic remembering.

Here, we develop an empirical approach to personal identity (in the described and original meaning; see [13]) based on a theoretical framework developed by Nozick [14]. We demonstrate how this framework—in contrast to other philosophical theories—assigns an explicit role to subjective evaluation and hence to empirical investigation of personal identity. We argue that this framework helps to overcome a tension identified in previous psychological approaches to personal identity, in which personal identity is studied as a type of similarity [5, 15, 16]: Following Nozick we assign a role to similarity metrics as prerequisites for judgments of continuity and place this judgment into a comprehensive theory of personal identity, in line with the theory of identity of objects, developed in [17]. This allows us to integrate existing psychological approaches as constructive elements of this theory and to resolve the identified tension by delineating the place for these elements. Our empirical paradigm allows for further systematic investigations into personal identity, which itself makes a constructive contribution to psychological research on personal identity, the self, future selves, and personal continuity.

## Personal identity and reidentification

This investigation into personal identity starts with a theoretical analysis of reidentification. Reidentification poses an important question concerning personal identity [1, 18]: Is $y$ at $t_2$ one and the same person as $x$ at $t_1$? We recognize and thereby reidentify acquaintances and connect their present state with earlier states, but we are also able to remember or imagine their existence in the past and the future. Furthermore, we each stand in a special relationship

to one person, our self, whose states in the past and assumed states in the future we can connect to our present state as belonging to the same person and nobody else [19]. Identity is inextricably linked to the self [20, 21].

Most of the cases of reidentification we encounter are benign and simple. At the age of 5, children can understand that the same individual can change over time without becoming someone else—for example, that animals grow larger over time [22]—and that reidentification "allows for a multitude of 'normal' changes" (see [21], p. 744). An older person is still reidentified by most to be the same person as their younger self, and people do not risk disrupting their identity when changing their spatial position by a few steps to the side. On the other hand, mere qualitative identity is not considered to be sufficient for reidentification [23, 24].

As the continuity of a person or object tends to be taken for granted in daily life, the metaphysical debate about personal identity and what it comprises has made extensive use of hypothesized and unusual circumstances. Philosophers often draw on thought experiments and counterfactual conditions to create puzzle cases for probing intuitions. These cases are meant to explore conditions under which sameness is preserved or disrupted. A famous identity puzzle is the ship of Theseus, described by Plutarch (*Vita Thesei*, 23.1) and taken up by Hobbes [25]: This putatively historical ship underwent constant change during which its wooden planks were exchanged one by one until its original material was completely replaced. Hobbes created a true puzzle case by imagining collecting the discarded planks and rebuilding of the vessel out of its original components. In this scenario two ships can claim to be the original, or numerically identical, ship ("eadem numero", p. 107), because both are continuers of the historical ship in a distinct sense: One is the continuation of the original ship in terms of the particles that made up the origina ship, the other in terms of its place in time and space. Intuitions amongst ancient and modern philosophers diverge on whether the spatiotemporally continuous vessel or the materially identical vessel should be reidentified as the original ship. In an experiment inspired by Plutarch, adults and children as young as 7 years old accepted that living organisms can undergo transformations and even lose parts without a loss of continuity: They chose an organism whose parts had been successively replaced over a second organism constructed out of the replaced parts as the original's continuer [26].

Investigations into personal identity have been similarly creative. Locke [9] chose the example of the soul of a prince finding itself in the body of the cobbler to ask whether this new person is a continuer of the prince or the cobbler. Similarly, thought experiments have tackled the question of personal reidentification in situations where more than one person can claim to be the continuer of an original person. The fission case [19, 27, 28] is a popular scenario for the study of the boundaries of identity ascription and preservation, in which a person (O) splits into two people (A and B) who stand in contentious relationship to the original person. Fission, which is assumed to occur in fictional teletransportation and duplication scenarios [6, 29, 30], creates two qualitatively different continuers, thereby making it possible to explore the dimensions involved in reidentification—for instance, whether spatiotemporal continuity needs to hold [14], or an appropriate causal connection between past events and current memories, actions and intentions, and persisting beliefs and desires [6, 30].

Moreover, facing more than one legitimate continuer poses a challenge to personal identity in a fundamental sense: The numerical identity relation is universally defined to be transitive [4, 6, 9, 13, 31]. If one object or person is the same as the second and the second the same as the third, the first must be the same as the third. In terms of the person O splitting into A and B, both A and B might be physically indistinguishable from O, and therefore separately might fulfill all the conditions necessary to be reidentified as O. Still, O cannot be numerically identical with both A and B, as A and B are not the same person and this would violate transitivity [6, 10, 17, 27, 32, 33] (as Locke [9] explained, "one thing cannot have two beginnings of

existence, nor two things one beginning, it being impossible for two things of the same kind to be or exist in the same instant, in the very same place, or one and the same thing in different places.", p. 204), and the question of whether O survives the split becomes contentious. This dilemma has led some philosophers to change their understanding of personal identity (e.g., [27]) and others to revise the concept of "person" in order to maintain transitivity in the problem case (e.g., [34]).

## The closest continuer theory

**The decision rule.** In our experimental approach to the concept of personal identity we are guided by Robert Nozick's [14] closest continuer theory, which allows us to connect debates in metaphysics on personal identity with several strands of research on identity in psychology. Nozick [14] offered a systematic approach to adjudicating between identity claims, which he called the "closest continuer view" (p. 34). The central claim is that if $x$ is an object or a person at $t_1$, then "something at $t_2$ is not the same entity as $x$ at $t_2$ if it is not $x$'s closest continuer" (p. 34); Nozick added that being the closest continuer was a necessary, but not a sufficient, condition. Some details need to be fleshed out, and for this purpose the decision procedure for determining whether an object continues to exist at $t_2$ or not is illustrated in Fig 1 as a fast-and-frugal decision tree [35]. Continuers of an object $x$ are named $y_1, y_2, \ldots, y_k$ and refer to the set of persons or objects with any form of claim to be the same object or person at $t_2$ as $x$ at $t_1$. The branches of the tree correspond to four different cases of interests, each with one of two decision outcomes.

If nothing at $t_2$ even qualifies to be a continuer of an object, then nothing at $t_2$ will be identical to $x$ and $x$ cannot exist at $t_2$ (Case 1). This is the first decision in the tree and possibly the simplest outcome. In the puzzle scenario of the Greek vessel introduced above, both the repaired and reconstructed ship are continuers of the original. Nozick argued that further decision steps depend on a metric of "closeness" to the original $x$, an intentionally vague term that will be further analyzed below. A second step tests whether there is any continuer that is close enough to qualify as being identical with $x$. If the ship catches fire and turns into ash at the same time that a swarm of fish happens to follow the ship's planned trajectory, no observer would identify the fish as the original ship; if this is the only possible continuer, the ship would cease to exist (Case 2). But both the repaired and the reconstructed ship could qualify as being close enough, so in this case a third question is required: Is there a single closest continuer at $t_2$? In the classic fission scenario [6], a person is copied particle by particle so that two persons are qualitatively identical and have more or less symmetric claims to be the original person. Determining whether there is in fact a closest continuer in this case requires a closeness measure and an ordinal relation defined over the set of its possible values. Small differences may matter, such as whether one of the continuers was a model for the other or whether one of them was created earlier than the other, which could be enough to determine a closest continuer (Case 4). If there is no such difference, or the differences are considered too trivial. Nozick [14] allowed for a semiorder [36] in conceding the perspective that a closest continuer "must decisively beat out the competition" (p. 40). Nozick would argue that it is "most plausible" (p. 63) to assume that the original no longer exists (Case 3): "If two things at $t_2$ tie in closeness to $x$ at $t_1$, then neither is the same entity as $x$" (see [14], p. 34). Hobbes [25] called the solution of having two numerically identical ships most absurd ("et habuissemus duas naves easdem numero; quod est absurdissimum", p. 107); the existence of two numerically identical persons at the same time would likewise violate the transitivity of the identity relation.

**The closeness metric.** The closest continuer view is intentionally formulated without a predefined metric of closeness, which gives it the character of a metatheory. Very different

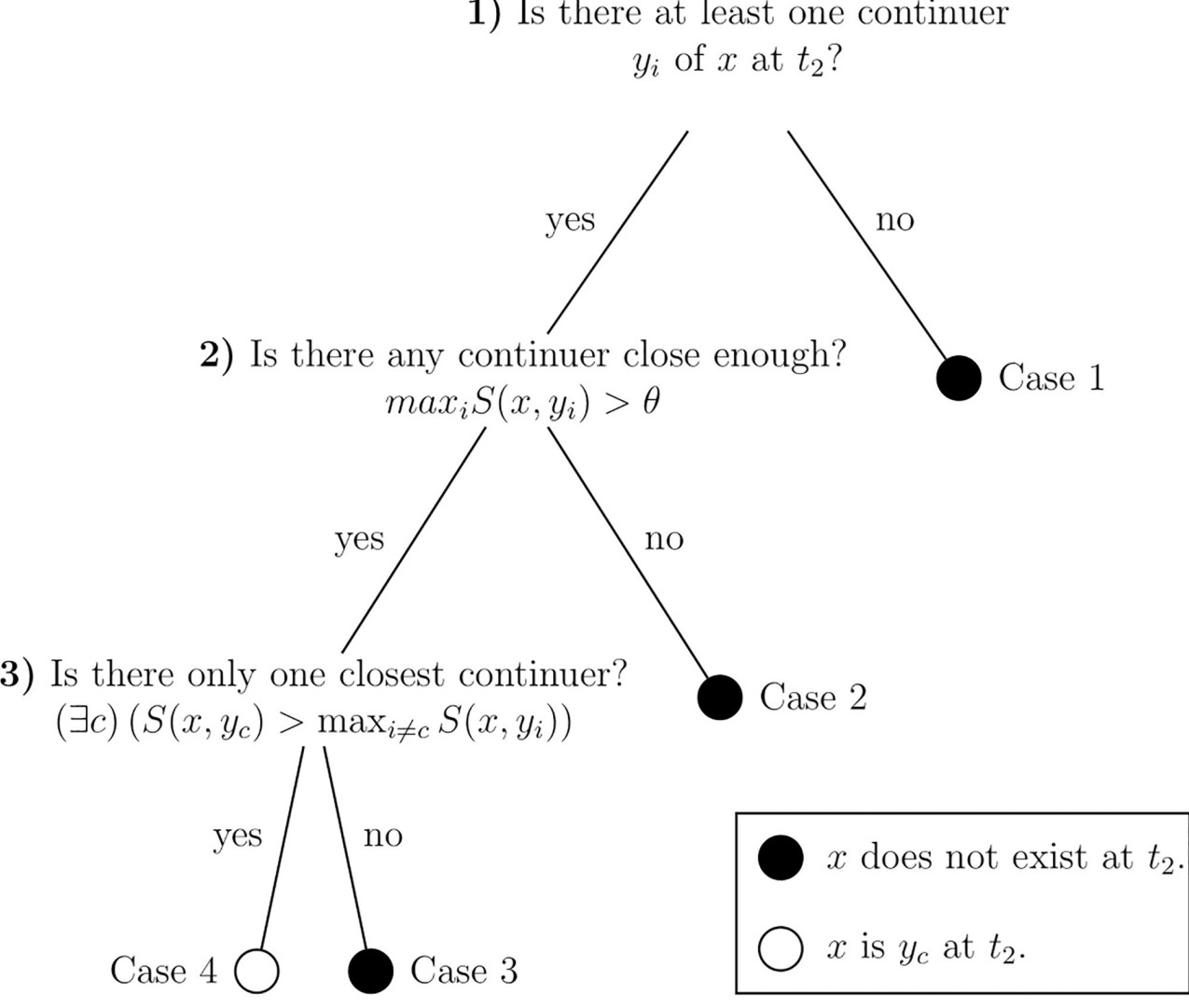

**Fig 1. Decision tree to determine the state of a person or object $x$ that exists at $t_1$: $Y_i$ is a possible continuer $i$ of $x$ at $t_2$ (adapted from Nozick, 1981).**

metaphysical approaches to identity could be reflected by the choice of different metrics. Closeness can come in degrees and therefore should not be conflated with numerical identity [5, 16, 37, 38]. While continuers are compared on a single dimension, Nozick conceives of this as a weighted additive model derived from multiple dimensions [17]. Notably, the metric chosen must reflect the type of entity that is compared—ships might be treated differently than stones, mountains, tennis rackets, and so on. Closeness does not reduce to mere similarity either, as causal knowledge can be invoked to disqualify continuers (see also [23]). While the identity of objects and persons both refer to sameness and continuity, first-person judgments of personal identity have implications for the judge who would not accept an arbitrary decision about what counts as herself (see [14], p. 34).

Instead of following "expert" opinions, Nozick argued that people "judge problem cases in accordance with the dimensions and weighting embodied in their own view of themselves" (p. 105). Whether this is considered inevitable or misguided depends on one's answer to the

question of whether there is a (metaphysically) correct view and correct weighting of dimensions and criteria. If so, variance from this one true view would indicate erroneous or at least ill-informed thinking. Nozick took a different stance, embracing the assumed variability of closeness metrics as expressing variability of what identity is for different persons:

> I suggest that there is not simply one correct measure of closeness for persons. Each person's own selection and weighting of dimensions enter into determining his own actual identity, not merely into his view of it. Because of our differing notions of closeness, for the same structural description of a problem case we can give different answers about which resulting person would be us, each answer correct.

(in [14], p. 106)

On this view, an empirical analysis of folk intuitions of closeness and personal identity is a *crucial complement* to metaphysical speculation. For related reasons, Mori [39] concluded that "a good enough definition of what personal identity in fact *is* may be provided by the way it is generally perceived, statistically speaking" (p. 278). Livengood and Machery [40] warned about the danger of philosophers assuming that their own intuitions can safely be ascribed to everybody else and suggest that "rather than guess from the comfort of your armchair, you ought to go out and check" (p. 126) whether decisions coincide or not [26]. Of course, potential variance in choosing and applying criteria might again be related to other individual differences [41, 42]. A good starting point for identifying a subjective closeness metric's dimensions is the concept people have of themselves, since changes to prominent aspects of one's self should impact the assessment of closeness between former and present self. Psychology has a long tradition of offering characterizations of what people regard as their self. William James [13] offered a rich definition, which serves as our starting point for investigating dimensions of closeness:

> *In its widest possible sense*, however, *a man's Self is the sum total of all that he CAN call his*, not only his body and his psychic powers, but his clothes and his house, his wife and children, his ancestors and friends, his reputation and works, his lands, and yacht and bank-account. All these things give him the same emotions. If they wax and prosper, he feels triumphant; if they dwindle and die away, he feels cast down, —not necessarily in the same degree for each thing, but in much the same way for all.

(p. 193)

Persons thus relate to themselves as comprised of "psychic powers," or psychology and memories; relations to other people; body; and possessions. The experiments we present study how participants sampled from the general population weight these dimensions of closeness to solve puzzle cases.

**Closeness, caring and survival.** Nozick suggested that the metric of continuer closeness was correlated with caring for the continuer. A metric of closeness features in a stream of psychological and economic research on the "future self" [43] that refers to it as connectedness and continuity, but often measures it with a single item, a choice between different visualizations of the predicted overlap between the present and the future self (10 years later). In these scenarios, only one continuer is assumed to exist in the future, but consistent with Nozick's suggestion, studies found perceived closeness to the future self to be correlated with reduced temporal discounting in experiments [43, 44] and more real-world financial savings [45], as well as a decrease in both cheating and the approval of unethical behavior [46]. Similarly,

anticipated changes to the self were found to increase the tendency to expedite rewards and delay burdens [47], as well as to increase charitable "selfless" giving [48]. In all these cases, however, only a single continuer is assumed to exist, which eliminates Case 3 as possible result from the decision heuristic. Changes in closeness to a single continuer only threaten identity if the decrease in closeness could reduce it below the threshold.

In a scenario with multiple continuers, however, Nozick argued that we should care about non-closest continuers "proportionally to their degree of closeness" (p. 64) and care disproportionately for our closest continuer. Molouki [49] provided positive evidence for this assumption by comparing allocation to others and the future self:

> Despite the finding that changes in the characteristics of liking, similarity, need, and deservingness seem to have a similar impact on resource allocations to others and to future selves, we nevertheless found a strong "premium" in future-self allocations that cannot be ignored.
>
> (p. 71)

When a person splits into two qualitatively identical persons, it is reasonable to argue that the original person is not numerically identical to either successor. On the other hand, it is not clear that the original person has died or did not survive the process, as either successor *alone* would seem to continue the life of the original. This opens the possibility that survival and personal identity might come apart after fission. Most philosophers would also agree that a minimum degree of closeness is also invoked in survival [6]. In the special case of multiple closest continuers, the present self may be psychologically connected to multiple identical future copies, at the same time. Each of these connections might even be based on the "right kind" of cause (and not based on pure coincidence or spurious connections). Parfit [6] referred to this relation (psychological connectedness with the right kind of cause) as "Relation R" (p. 262) and believed with others (e.g., [10, 50, 51]) that not identity but Relation R mattered in survival. He assumed both relations to be aligned in ordinary cases, but argued that only identity was constrained by transitivity. There is no contradiction of someone being close to one and only one of two persons that are close to each other, just as it is possible for the eye to distinguish a specific color from one and only one of two other colors that are as a pair indistinguishable to the human eye [36]. Faced with two closest continuers, I cannot be identical to either one, on many accounts, but I might still survive [6, 10, 50, 51]. I would be entitled (before fission occurs) to care for these continuers but mistaken in regarding either or both of them as myself.

Survival should not be confused with mere existence; it is not enough that there are continuers. Non-survival could be regarded as corresponding to Cases 1 and 2 in Nozick's decision scheme: If there are no continuers, or there is no continuer close enough (above a necessary threshold that may or may not coincide with the one used for identity assessments) then the person does not survive. A second focus of our empirical investigation is therefore to find out how participants resolve these tensions between identity, closeness, and survival. We will now describe the scenario used in our experiments and develop our research questions and hypotheses.

## Experimental scenario

We investigated attitudes and intuitions regarding personal identity and reidentification with participants from the general population recruited via Amazon Mechanical Turk. They were presented with a variation of fictional fission scenarios devised by Williams [28] and Parfit [6] in two studies from the first-person perspective. The full description of experimental details

and instructions are presented in S1 Supporting Information (sections 1.1. and 1.2.). In both studies, participants were told to imagine that they were about to be split into two continuers [17, 52] who would each inherit different dimensions of attributes from the original person (similar to [53], Exp. 3 and Exp. 4). Guided by James [13] and drawing on the theoretical and experimental literature, we decided on five dimensions for determining closeness between the participants and their continuers: (a) body and appearance, (b) personality and psychology, (c) memory and knowledge, (d) friends, and (e) possessions. Participants read the following scenario:

> *In the future, hyperspace travel has become normal. You enter hyperspace to travel large distances and leave it at your new destination. Unfortunately, the technology still has some problems. One rare incident occurs while you are traveling from one planet to another planet: For a brief moment, the present universe overlaps with a parallel universe. Your travel agency contacts you while you are still in hyperspace and informs you that due to the overlap it has been calculated that, unfortunately, not one but two people will leave the hyperspace at your target destination: person A and person B, while you will no longer exist in your present state.*

The composition of the continuers was systematically varied, with each of the five original dimensions being assigned to just one continuer. For example, participants were told that "one of the two persons has your exact body and appearance, the other person has the body and appearance of a randomly chosen person of the same age and gender." A colored table demonstrated the assignment of original components to persons A and B (see Fig 2). All dimensions not inherited from the original person were said to be copied from a randomly chosen person of the same age and gender as the participant. Consequently, both continuers qualified as fully functioning and complete persons (compare [15]), and both were parallel products of the same incident. Participants in Study 1 were asked to split a sum of money between the two persons before fission occurred:

> *The travel agency's insurance company is willing to pay $100 000 to compensate you for the problems caused by the incident. Fortunately, they can contact you before you leave hyperspace. You have to decide now how to distribute the money between the two people that will exist (in your place) after you leave hyperspace. (The values you choose must add up to 100,000.)*

This situation bears some resemblance to the dictator game in behavioral economics [54], in which a "dictator" divides up a sum of money between themselves and a passive recipient.

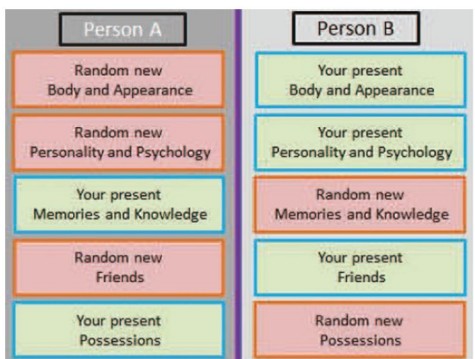

**Fig 2. Examples of tables shown to participants in Study 1 (left) and Study 2 (right) to characterize the scenario condition.**

The dictator game has been used to measure deviations from behavior guided by pure self-interest. Typically, in adult Western populations, the average amount given to the second party is smaller than the amount kept [55], though some players split the amount equally [56]. For McMahan [57], the "rational degree of egoistic concern" should vary with the degree of connectedness and continuity. In our scenario, the direction of self-interest should be determined by the perceived closeness to the continuers [6, 43, 45, 48]. Prudential concern will only be shown towards someone believed to be oneself [58]. Participants should therefore favor the continuer judged to be the closest continuer, as "future selves are *not* treated like others when it comes to the absolute amount allocated to them" (in [49], p. 71). The difference in caring should also track the difference in closeness, as "it can be rational to care less, when one of the grounds for caring will hold to a lesser degree" (in [6], p. 313). In the extreme case in which even the continuer a participant considers to be their future self does not appear closer than a "stranger" [43, 59], this should result in equal splits if they apply a strict fairness allocation principle (this may come close to Case 2 in Fig 1). To downplay the importance of fairness considerations, we did not use windfall gains or "house money" in our scenario, but specified that the continuer was entitled to compensation from an insurance company for the fission accident [60], linking the future compensation to the person's past suffering and restricting the entitlement therefore to the future self [12, 61]. In our design, all possible composition patterns were presented twice, with the continuers assigned to the two positions switched in the second version. Thus, any effect due to fairness consideration would move monetary allocations towards an equal split, effects unrelated to the factorial distribution should stochastically even out. Responses influenced by these two factors could thus attenuate discovered relationships, but they should not introduce spurious new relationships.

To explore interindividual variation in weights, the scenario was complemented by questions concerning how participants separately judged the importance of the varied dimensions in identity and survival. In addition, we collected summary judgments regarding the state of the original person after the incident. Specifically, we asked whether the original person that underwent fission was judged to survive this incident, ceased to exist after this incident, or was considered to be two people at the same time after this incident. These questions probe whether participants' intuitions correspond to Nozick's scheme and connect to a stream of research in psychology and experimental philosophy.

## Research questions and hypotheses

**Dimensions of assessing closeness.** The closeness metric is the centerpiece of Nozick's closest continuer theory. This metric reflects which dimensions with which weight determine decisions between continuers in cases of conflict. It therefore specifies what exactly people refer to as their future selves and reflects what people deem to be their essential parts and how they judge connectedness over time. Here we address an open question in the theory, namely what subcomponents with which relative weights are part of the subjective similarity metric (see [14], p. 69). In the following, we explain our choice of features and develop hypotheses regarding the monetary allocation task in Study 1. Each feature is discussed in turn. Note that our design allows for each feature to receive a positive weight in decision making about allocations, even if splitting positively evaluated components can create conflicts in specific scenarios. Note that we do not derive these features from Nozick's theory, as the closeness metric could accommodate all or none of them. Thus, our hypotheses are derived from the cited theories in philosophy and psychology.

**Body and appearance.** Body and appearance are the most visible attributes of a person. If the primary concern is one of spatiotemporal continuity, changes in shape and form endanger

both survival and identity. In addition, a person spending substantial time on their body and appearance (such as body-builders) will likely consider the result of their work an essential part of their self. The continuation of the body seems to be an obvious place to look for personal continuity. A reductionist would see any other feature of a person as entirely dependent on the body. Animalism [62] holds that each human is numerically identical to one animal in one body; the brain is treated as only one organ among many. Yet the strict body criterion has fallen into disrespect with most modern philosophers [11], even when embodied approaches regard mental life as severely dependent on the physical body [63]. Also, tracking people based on visual appearance only has been proven unreliable [1]—think of the multiple villagers who confirmed that du Tilh was Guerre. On the other hand, participants might not share this perspective, and Meyer and colleagues [64] show that participants on Mechanical Turk expressed the belief that mere transplants of body parts might confer non-bodily aspects of the donor. We therefore hypothesized that participants would view their body and appearance as important in determining which of the two continuers deserves compensation, and—ceteris paribus—would allocate more money to the continuer that inherited their body as opposed to the continuer with a randomly selected body. This hypothesis corresponds to expecting a main effect for the inheritance of the original body across scenarios:

**Hypothesis 1a** *Participants will allocate more money to the continuer inheriting their body and appearance.*

**Personality and psychology.** People's personality and psychology are a central part of how they behave and react to the world. In a sense, "personality" is what makes a person different from or similar to others. In prominent theories of personal identity, psychological factors are considered the most important criterion for reidentification [12, 15] or self-persistence [31]. [65] demonstrated that participants rated a loss of moral attributes as having a larger impact on perceived change than a loss of other personal attributes. Parfit [6] and Campbell [66] stressed that what matters in survival is psychological connectedness between former and future self, which led us to the following hypothesis:

**Hypothesis 1b** *Participants will allocate more money to the continuer inheriting their personality and psychology.*

**Memory and knowledge.** Memory and knowledge are central components situating a person in the environment and linking their present state to a personal past. Since Locke [9], autobiographic memory has been considered a key part of how people make sense of or even construct their own self-reference [67], which is equated by some with a psychological notion of identity. Knowledge, on the other hand, includes items in memory that are functional in making people succeed in their endeavors. Like body shape and appearance, knowledge can be the result of intentional care and self-modification. Transhumanists consider the possibility of uploading the "contents of the brain to a different, more flexible medium" (in [68], p. 74) as a way of prolonging a person's existence. Memory and knowledge are of premium importance in any information-theoretic or cognitive-agent perspective on humans. It should be stressed that not all memories seem to be descriptively accurate and related to objective matters of fact. Therefore, it should not be considered a "logical truth" [69] that an act of remembering an event necessarily involves having witnessed it. Indeed, empirical studies on episodic memory demonstrate remembering to be more than "a simple connection between discrete moments of consciousness" [70]. Children faced with staged duplication scenarios of hamsters [71] assumed that fewer mental than bodily properties would be conferred to the copy; stressing the identity of the original hamster by naming it further decreased children's expectations that a hamster's episodic memories would be transferred to its copies. The importance of memory for identity might be enhanced by the role of "signature stories" [72] that acquire self-defining functions (see [73]). In this view, memory is understood as more than a passive, digital storage

facility [72], as stories that are told and retold can be reinterpreted in self-narration [74]. Some authors refer to identity based on self-narration as "narrative identity" [32], as opposed to numerical identity. Memories and knowledge about the self appear to have a privileged status: They tend to be better protected in semantic dementia than other types of memories [20]. The following hypothesis reflects the privileged nature of memory as a basis for identity:

**Hypothesis 1c** *Participants will allocate more money to the continuer inheriting their memory and knowledge.*

**Friends.** Sociological and social-psychological theories of the self and the person [75–78] emphasize interactions and relationships. If a person's place in society is shaped by social responses, claims, responsibilities, and roles, then their personal identity cannot be easily separated from the identities of people with whom they are socially connected. [79] argued that "[a] person has identity only in relation to other people and other roles" (p. 49). Participants in a study by [15] spontaneously generated terms describing linkages to others when asked to list characteristics of persons. Many social psychologists treat people's identification with groups and self-categorizations in terms of social categories [80]. These categories are validated by others [81] and play a prominent part in constructing the self [82]. If the self is shaped by the eye or the mere existence of the beholder, a change in the social setting (e.g., replacement of all friends, as close and self-selected components of the personal social network) can be expected to have a major impact on identity.

Participants in our experiments may have been exposed to multiple discourses about identity. According to Davis [83] (p. 373), social psychologists contrast the concepts of (potentially multiple) social identities with the notion of personal identity as "an individual's identity *apart* from others," whereas philosophers view personal identity as numerical identity. It is still worth exploring whether participants consider the continuity (or the continuous evolution) of the personal network as a criterion to determine identity between persons in the scenario. After all, they might share the intuitions of social psychologists [80] more than those of philosophers, who put less emphasis on social roles. When imagining possible future selves, participants from different cultures focused not only on abstract self-improvement, but also on future social relations [84]. Considering the continuity of social relations as a criterion for personal identity led us to the following prediction:

**Hypothesis 1d** *Participants will allocate more money to the continuer inheriting their friends.*

**Possessions.** Finally, several theories address the interaction between internal human processes and outside objects. These theories raise the possibility that classic boundaries and distinctions might be mistaken, or at least lacking in precision. From the extended mind perspective [85], internal cognitive processes can interact with external structures and processes to extend the boundaries of cognitive systems [86]. For example, if someone relies on technological support to maintain memories and organize their intake of new information, that technology becomes intermeshed with their internal faculties.

Following from [13], marketing scientists have transferred a related concept of extension to the notion of an "extended self" [87]. In an analysis of personal self-concepts derived from James's list of features of the self, Prelinger [88] found that "possessions and productions" were ranked as less central than body or psychological and personally identifying characteristics, but as more central than the external environment or other people. Referring to these results, the extended self approach [87] considers consumers to literally be what they possess, with a potential for self-extension through control over those objects. Diesendruck and Perez [89] observed that even children do not clearly separate objects from the moral qualities of their owner and may interpret objects as "physical manifestations" of a person's characteristics: "we become the objects that we own" (p. 11). People's experiences of themselves and their day-

to-day lives becomes interlinked with the objects they own: They actively use objects and brands to construct and clarify who they are [90, 91], and losing or replacing possessions can impact their perception of themselves. From this point of view, possessions are "intimately tied to [a person's] sense of self" (in [92], p. 550). LeBarr and Shedden [93] recently demonstrated that induced ownership leads new owners of objects to form associations between the owned objects and self within minutes.

Possessions play a defining role in a person's sense of self and personal identity. This could stem either from the fact that people's cognitive processes rely on outside objects (e.g., external storage or tailored aids), or from the significance those objects have acquired in defining who the person is. People might therefore consider the continuity of possessions in determining identity. We therefore formulated the following prediction:

**Hypothesis 1e** *Participants will allocate more money to the continuer inheriting their possessions.*

In theory, all five hypotheses could be true at the same time, this would correspond to five main effects across all scenarios, in which all possible combinations of original features are explored.

**Monetary allocation and personal reidentification.**    The questions concerning the state of the original person after fission allow us to study participants' decision making regarding the consequence for personal identity and survival. We assume that the allocation of money tracks personal reidentification and not other-regarding concerns. We ask two questions in the post-questionnaire of the form "You are person $X$ after the incident" (all questions are listed in S1 Supporting Information) and expect that answers to these questions will respond to scenario conditions in the same way as monetary allocations:

**Hypothesis 2** *Participants' personal reidentification with either continuer will follow the same pattern across conditions as the monetary allocation.*

**Explicit weighting of dimensions.**    The weighting of dimensions is assumed by Nozick [14] to be an implicit process. Participants might therefore not have access to the exact weights that would make it possible to predict their decision (and decisions in all other possible scenarios). Nonetheless, it seems plausible to assume that participants can identify important and less important dimensions. When asked about the importance of the five dimensions (body and appearance, personality and psychology, memory and knowledge, friends, possessions) for determining the identity between persons, participants' weightings should on average correspond to the identified importance for allocation decisions. Asking for the identity "between people" ensures that participants do not answer the question with one of the divergent nonrelational meanings of personal identity commonly used in psychology in mind (see also the General Discussion).

**Hypothesis 3** *Participants' subjectively assessed importance of dimensions will correspond to the importance of factors for the allocation decision.*

It has been argued that the closeness metric drives reidentification decisions in the fission case. As discussed above, Parfit [6] and others claimed that it is not the identity relation that matters in survival. Rather, he proposed that what matters is Relation R—psychological connectedness and the right kind of continuity—meaning that hypothetically, there can be cases of survival without the preservation of identity. But if connectedness plays a central role for survival judgments, then intuitions about survival should be influenced by a closeness metric, but not necessarily the same metric as identity judgments. To test whether the same or different weights are applied, we also asked participants to assess the importance of the five dimensions in determining whether a person survives. As we do not have a strong prediction, we formulate this as a research question:

**Research Question 1** *Is the importance assigned to the five dimensions the same for determining survival and the identity between people?*

Philosophers differ in the degree of interindividual variance they would expect in the perceived weight of dimensions. While Nozick [14] entitles observers to idiosyncratic weighting, others argue for universally shared intuitions or at least universally shared normatively correct intuitions that should also constrain variations in weights.

**Personal reidentification after fission.** Following Nozicks's [14] decision scheme, our scenarios might correspond to one of three cases for each participant (assuming that the two people after the incident qualify as continuers). Neither continuer can be close enough (Case 2), one can be close enough and closer than the other (Case 4), or both can be close enough but appear too similar to decide between them (Case 3). Case 3 can occur even if continuers in our scenario are never identical. The questions asked after the main task allow us to gain more insight into the decision-making process and the reasoning behind the allocation decision. Of course, no empirical finding could be interpreted as a refutation of Nozick's normative theory. The analysis will thus focus on the usefulness of this framework for organizing empirical responses by (mostly) non-philosophers. Thus, we formulate research questions, and not hypotheses. In a first step, answers to the open question about reasons for the allocation will be analyzed for correspondence with the closest continuer theory:

**Research Question 2** *Is there evidence in open answers for participants following the logic of the closest continuer theory?*

There is near-universal agreement in philosophy that declaring two distinct continuers to be the same person as the original is inconsistent and clashes with the assumed transitivity of identity. In experiments with non-philosophers, however, psychologists have found evidence for violations of transitivity in people's assessment of identity relations. Rips and colleagues [17] found that faced with two continuers of animals made up of particles from the original, most participants saw both of them simultaneously as the original, with similar results for objects such as possible continuers of icebergs. The authors excluded simple explanations due to response formats, and Rips [33] replicated these results under several variations. Weaver and Turri [52] claim to be the first to show that these results extend to judgments about the identity of persons. Specifically, they demonstrate that participants violate the "one-person-one-place" rule (p. 34) after a transporter incident duplicates the traveler and locates the person in two places at once in several variations of the scenario. The authors wonder whether "perhaps drastic psychological change across co-located figures could lead people to judge that the figures [. . .] are different persons." (p. 35). Our scenario allows us to explore this question by analyzing the frequency of participants affirming that they are two people after the incident and the conditions under which these responses are made:

**Research Question 3** *Are there conditions under which participants consider the original person to be two persons after the incident?*

If reidentification judgments are determined by a single dimension, existence should be guaranteed under all scenarios, as all original dimensions are present in one of the two continuers after the incident. If the persistence of two or more dimensions is considered to be necessary for survival, then splitting these dimensions would result in a loss of existence. The closest continuer scheme would predict nonexistence responses in cases where no single continuer is close enough. In this scheme, nonexistence is also predicted if two dimensions are separately sufficient for continuity but both continuers are too similar in closeness. We therefore include a final research question:

**Research Question 4** *Are there participants who consider the original person to be nonexistent after fission?*

Again, the same scenario might be interpreted in different ways by different groups of participants and we will investigate both stability and minority perspectives.

## Study 1

### Methods

**Experimental setting and sample.** Participants were recruited via Amazon Mechanical Turk. Our personal identity questionnaire was embedded in a Human Intelligence Task (HIT) containing a series of cognitive and evaluation tasks lasting an expected 45–60 minutes. Participants were recruited for this task over a period of about 100 hours. A minimum approval-rating filter [94] of 95% was combined with attention-check questions at the beginning of the survey to ensure sufficient attention to questions and instructions. We report on all participants who passed these checks and described themselves as U.S. citizens. Data collection was not restricted to U.S citizens, as other parts of the survey—data supporting [95]—required a broader sample, but a large majority (75.8%) of the respondents were Americans (which is typical for MTurk; see [96]). U.S. participants are the largest group of workers on the Mechanical Turk platform. They are regarded as less representative of the population than national probability samples but as more representative than students or convenience samples [96, 97]. There were 196 responses from India and 29 from other countries. Both samples were too small for separate analysis and—relevant to the complex scenario used in our experiment—there were concerns about the English comprehension and data quality of Indian participants [98, 99].

A total of $N = 704$ responses were analyzed in Study 1. The mean participant age was 33.4 years ($SD = 10.7$ years) and 49.7% of participants were female. Participants who completed the survey were remunerated with a fixed payment of $2.50 and a variable bonus of up to $1.50, contingent on their performance in the HIT. Our survey was always embedded at the same position within the HIT. The survey was coded and conducted using the Qualtrics platform. The study received IRB approval (HEC Lausanne Ethics Committee), and participants gave electronic consent after reading a consent form prior to beginning the study.

**Experimental design.** We recruited a large sample of participants in a five-factorial between-subjects design with single scenarios to avoid potential order effects [100]. Each participant was exposed to one of 32 variants of the scenario that were created by the $2^5$ possible distributions of the five features between the two resulting persons. The 32 scenario variants in the $2 \times 2 \times 2 \times 2 \times 2$ between-subjects factorial design were identical, with the exception of the diagram shown to illustrate the division of attributes after the accident (see Fig 2). This complete five-factorial design balances out any positional preference, as each split between the two persons recurs with the positions reversed. In addition, any preference for equal splitting that disregards personal attributes should be expressed equally in all conditions without distorting the results. Each participant was presented one variant of the scenario. We thus obtained independent data for 32 variants differing the allocation of the five features to the two resulting persons. Having read the scenario description, participants were asked 18 questions on the scenario. First, they were instructed to distribute a sum of $100,000 in insurance money between the two resulting persons using two sliders with a fixed-sum constraint. They were then asked to explain their choice of distribution in an open-answer format. Next, participants responded to a series of statements regarding identity and survival after the accident on a continuous slider scale (from "strongly disagree" to "strongly agree"). Finally, they judged the importance of the five features separately, once for the determination of identity, and once for the determination of survival, on a similar continuous scale (from "not at all important" to "extremely important").

## Results

**Allocation of insurance money.** Allocations to person A and person B spanned the full possible range (see Fig F in S1 Supporting Information). The mode of the distribution was an equal split between A and B, with other participants choosing amounts very close to $50,000. Few participants chose extreme splits; more respondents chose values around multiples of $10,000. We computed an ANOVA with five between-subjects factors (presence/absence of the five original features in person B) and the insurance money allocation to person B as criterion (see Table A in S1 Supporting Information, for the complete results). All possible interaction terms were included in the model. Cell counts varied from 15 to 30, with a mean of 22.0 participants ($SD$ = 3.5). For the results presented below, the combined dimensions psychology/personality, memory/knowledge, and body/appearance will be referenced as psychology, memory, and body, respectively.

We found two significant main effects: for psychology ($F(1, 672) = 45.01$, $p < .001$, partial $\eta^2 = .06$) and memory ($F(1, 672) = 45.20$, $p < .001$, partial $\eta^2 = .06$). None of the other main effects were significant; neither were any of the 26 interactions. The two main effects remained significant after Bonferroni correction for the family-wise error rate. In addition, these effects appeared to be robust across all possible distributions of the other three components (see section 2.1.2. in S1 Supporting Information). Our findings on the allocation of insurance money thus support Hypothesis 1b and Hypothesis 1c (participants allocated more money to the continuer inheriting their psychology and personality, and their memories and knowledge). There was little or no support for Hypothesis 1a (body and appearance), Hypothesis 1d (friends), and Hypothesis 1e (possessions).

**Identification with continuers.** Participants responded to six statements about their perceived state after the incident. Two questions probed their identification with person A and B, respectively. The responses to these two questions offer an alternative route to analyzing the influence of the location of the five features.

We calculated two ANOVAs (see Table B and Table C in S1 Supporting Information), with the five between-subjects factors and the two statements describing identification with person A or person B as the respective criterion (as identification—unlike monetary allocation—did not have a fixed-sum constraint, the second ANOVA is not redundant). In both ANOVAs, we observed a significant main effect of psychology ($F(1, 660) = 70.26$, $p < .001$, partial $\eta^2 = .10$ for person A and $F(1, 662) = 44.83$, $p < .001$, partial $\eta^2 = .06$ for person B) and memory ($F(1, 660) = 76.50$, $p < .001$, partial $\eta^2 = .10$ and $F(1, 662) = 59.32$, $p < .001$, partial $\eta^2 = .08$). In both ANOVAs, parameter signs indicated that the person with each respective component received more money. The only other significant effect was a weak three-way interaction between body, friends, and possessions ($F(1, 662) = 8.69$, $p = .003$, partial $\eta^2 = .01$) for person B. This result is difficult to interpret and may be due to sampling error (the corresponding effect for person A was virtually nonexistent ($F(1, 660) = .15$, $p = .70$, partial $\eta^2 = .00$). The pattern of results is thus very similar to the one obtained for monetary allocations; in other words, the data support Hypothesis 2 (participants' personal reidentification with either continuer followed the same pattern across conditions as the monetary allocation).

**Explicit weighting of dimensions.** Fig 3 summarizes participants' answers to the 10 questions assessing how they perceived each of the five dimensions to be related to identity and survival, respectively.

The five dimensions were gauged to be of similar importance for identity and survival; in addition, answers pertaining to the same dimension were highly correlated (the full intercorrelation matrix is included Table E in S1 Supporting Information). Friends were judged to be slightly more important for identity than for survival, on average. Corresponding to the main

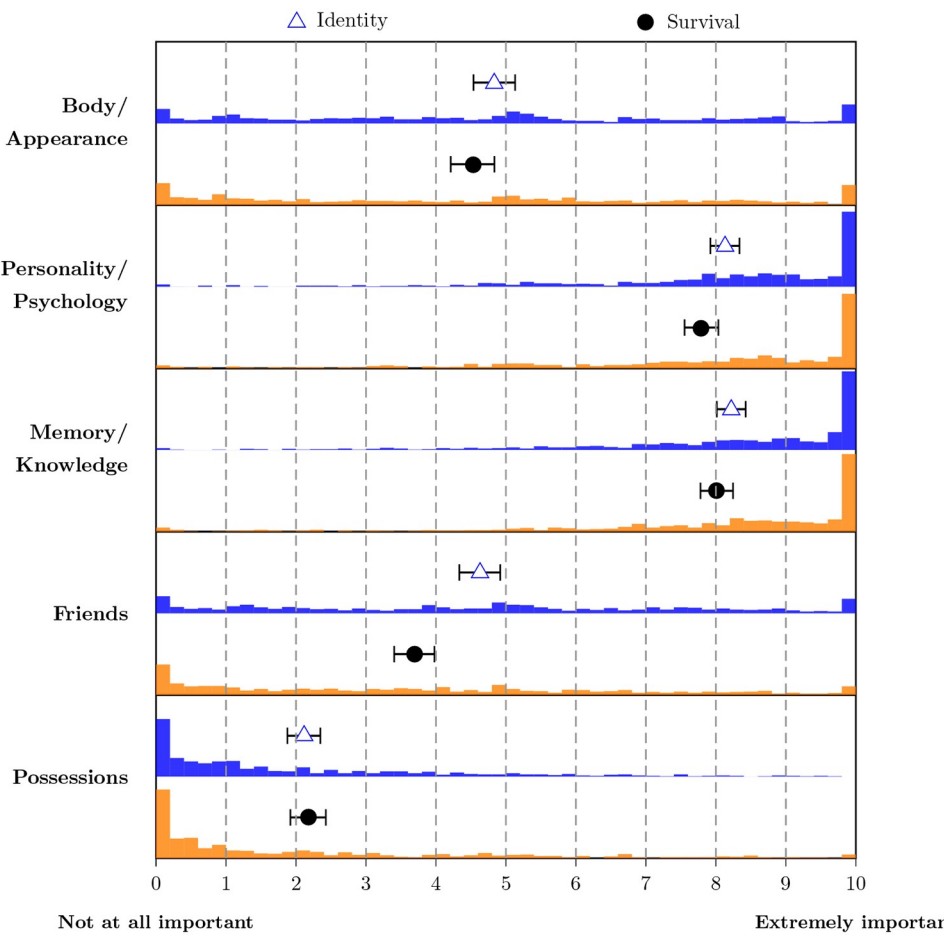

**Fig 3. Importance assigned to the five dimensions for identity and survival (*N* = 704).** Triangles denote the means and blue bars show the histogram for answers to the five identity questions; dots and orange bars show the corresponding information for the survival questions. Whiskers signify the 99% confidence intervals for the means.

effect of the ANOVA reported above, psychology and memory were considered more important than the other three dimensions, again supporting Hypothesis 1b (psychology) and Hypothesis 1c (memory), and given the similar pattern to the ANOVA results, supporting Hypothesis 3 (participants' subjectively assessed importance of dimensions corresponded to the importance of factors for the allocation decision). In contrast to the previous analysis, body and friends were judged to be of medium to low importance, while only possessions were regarded as unimportant. A closer analysis of the distribution of responses shows that responses for both body and friends spanned the full range of values and were roughly symmetric around the mean. As participants did not make a substantial difference between identity and survival, we can give an affirmative answer to Research Question 1: The importance assigned to each of the five dimensions was similar for identity and survival. In S1 Supporting Information, we explore the connection between the dimensions' judged importance and their monetary evaluation (see sections 2.1.7. and 2.1.8.).

Given the complete individual explicit ratings of all five conditions, we also searched for subgroups of participants with divergent weighting patterns. We conducted a cluster analysis on the matrix of 10 weightings for all participants (see section 2.1.9. in S1 Supporting Information). The resulting final six cluster centers demonstrate unanimity in the highest ranking of

psychology and memory. Only in one cluster were possessions rated somewhat important (along with all other dimensions). The clusters were equally split into considering body relatively important and relatively unimportant, and divided similarly for friends, with two clusters regarding both to be important. These results could be considered as qualified evidence for Hypotheses 1a (body) and 1d (friends), but, again, there is little evidence for Hypothesis 1e (possessions).

**Personal reidentification after fission.** The qualitative answers of 521 participants (74%) could be interpreted as reasonable reactions to the encountered scenario; other answers show some sources of confusion or alternative interpretations. For the largest subgroup of 87 participants, their allocation was targeted at mitigating losses, which runs counter to the instructions. The overall pattern indicates a positive answer to Research Question 2: Participants explain their monetary allocation often in reference to closeness or identity relations with continuers. Some of the reasoning involves a direct comparison between continuers, declaring one closer, closest or having "more" of the original person. Participants who allocate close to equal splits claim to be indifferent between continuers, or describe them as deserving equal compensation or payments. Some invoke fairness towards both continuers, which could either be consistent with the closest-continuer theory in that participants see both continuers as other people or indicate an influence of other-regarding preferences not considered in the theory. A complete analysis of participants' qualitative answers is given in S1 Supporting Information (section 2.3.).

Fig 4 summarizes the responses to the six post-questionnaire items. As the characteristics of person A and person B changed between experimental conditions, we show the maximum and minimum value for each participant (in six cases, one answer was missing; we considered the remaining answer as both minimum and maximum). Participants who identified with one

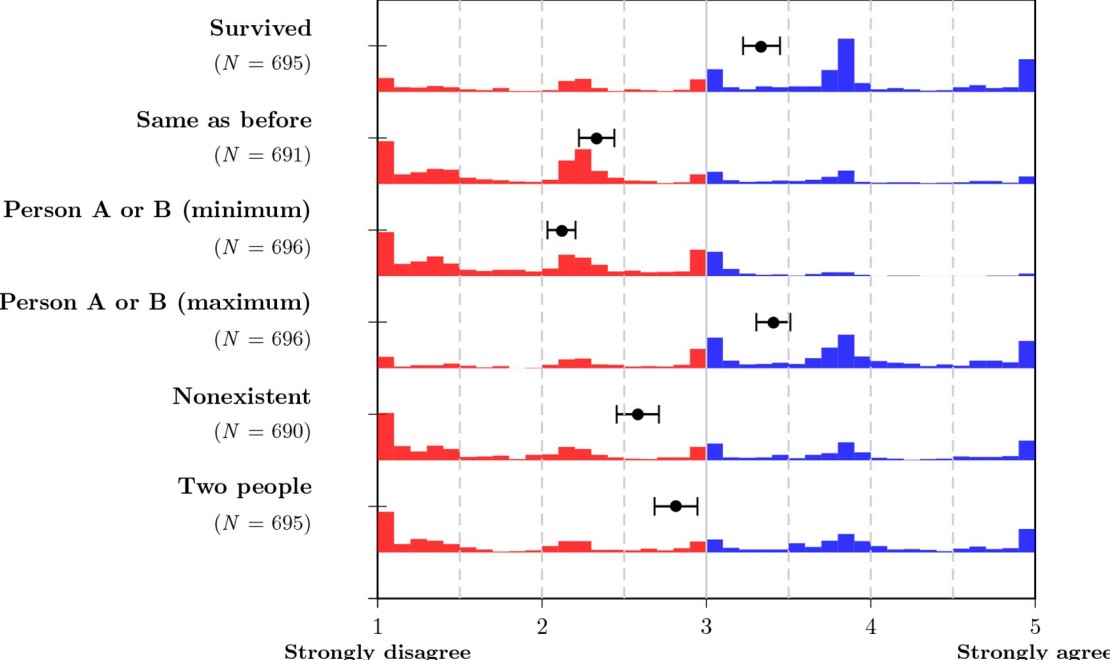

**Fig 4. Histogram and means of responses to post-questionnaire items.** Blue bars indicate a tendency to agree with the statement; red bars, to disagree. For the two questions regarding being person A or person B (whose composition differs between conditions), the smaller and the larger value for each participant were pooled and analyzed as minimum and maximum value. Dots represent means; whiskers denote 99% confidence intervals around the means.

continuer should express a high degree of identification with one person and a low degree of identification with the other. A high minimum indicates a high degree of identification with both continuers; a low maximum, a low degree of identification with both.

An analysis of all six response distributions revealed that, on average, none of the statements was strongly endorsed or strongly rejected. Participants tended to affirm that they had survived the incident, and identified to some degree with one (but not the other) of the resulting persons. This maximum identification was positively correlated with the maximum percentage of dimensions retained in one of the two persons ($r = .103$, $N = 696$, $p = .007$) and the maximum of the two monetary amounts allocated ($r = .265$, $N = 696$, $p < .001$). On average, participants tended to reject the statement that they were the same person before and after the incident.

Statements suggesting nonexistence or being two persons after the incident were both rejected on average, but the mean response was closer to a position of indifference. On average, participants agreed less with the statement that they were two persons after the accident when memory and psychology were united in one person. An ANOVA with identification with both continuers as criterion showed a significant interaction between the location of memory and psychology ($F(1, 663) = 28.43$, $p < .001$, partial $\eta^2 = .04$, see Table D and Fig H in S1 Supporting Information). All averages were close to indifference in all conditions. As the histogram in Fig 4 reveals, this average indifference hides polarized answers, with some participants agreeing strongly and more participants disagreeing strongly.

The monetary allocations also revealed differences between conditions in which memory and psychology components were united versus those in which they were split, as Fig 5 illustrates: When memory and psychology were united, the differences in allocations were larger;

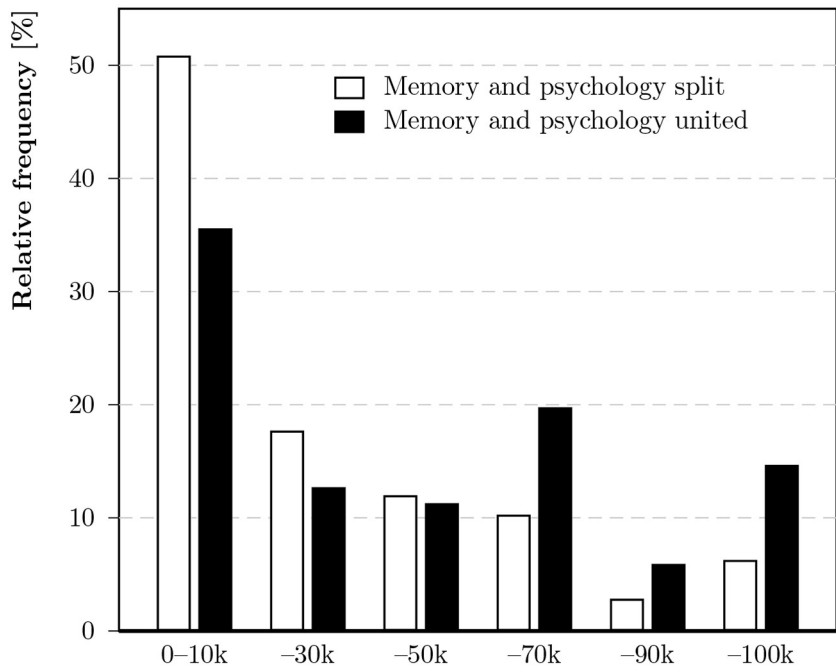

**Fig 5. Relative frequency of absolute allocation differences, separately for conditions in which memory and psychology were split vs. united (intervals include the upper limit).**

when they were split, the differences were smaller (more than 50% of participants split the money roughly equally in this condition).

Research Questions 3 and 4 aim at judgments violating transitivity by either identifying with both continuers (Research Question 3) or neither continuer (Research Question 4). There was mixed support for the idea that people violate transitivity in their judgments of identity in either fashion. The S1 Supporting Information (see sections 2.1.1., 2.1.4.–2.1.6.) offers further insight into how the location of psychology and memory influenced participants' identification with either continuer. The aggregate level results followed the pattern of the ANOVA findings: Identification (and allocation) was higher when memory or psychology continued, particularly when the continuity of both was preserved. Nevertheless, the individual analysis shows some participants identifying with both continuers at the same time and other participants identifying with neither. A second cluster analysis (see section 2.1.10. in S1 Supporting Information) revealed clusters of participants corresponding in their responses to those expected in the relevant three cases in Nozick's decision scheme (there are always two possible continuers, so Case 1 cannot occur). We found clusters that endorse being two people, being nonexistent, and also being both nonexistent and two people after fission. Cluster membership is significantly related to features of the scenario and monetary splits. Most responses can be interpreted meaningfully through the lens of the closest continuer theory, but we observe interindividual variability and potentially divergent concepts of persons.

## Discussion

The results of Study 1 indicate that the closest continuer theory is a productive framework for studying the understanding of personal closeness and identity by participants. On the aggregate, the espoused theory is best described as putting a large weight on psychology and memory (which corresponds to using a psychological criterion of identity): Memory and psychology are considered to be more important than the other three dimensions, with more money being allocated to continuers who possess the original attributes. Similarly, the average identification with continuers depends on the distribution of memory and psychology. Only when moving to the individual level do we find evidence for the importance of social and material dimensions: Friends and the body were seen as important by subgroups of participants, but there is very little evidence for considering possessions as an important component in the closeness metric. This is consistent with the relative weightings reported in [88]. While the aggregate results therefore conform with some of the most established criterial theories of identity [6], the presence of interindividual variation gives *prima facie* some vindication to Nozick's [14] contention that people do not share a single closeness metric. At the same time, we find additional evidence for the violation of the transitivity norm in personal identity judgments. Not most, but still numerous participants endorsed the view that two people were equivalent to the original after the accident (we return to this point in the general discussion).

Study 1 had a number of conceptual and methodological limitations, however. First, answers to the allocation question seem to have been influenced by fairness considerations, as indicated by open answers and the high percentage of equal-split allocations. In a meta-analysis of 83 dictator-game studies, Engel [56] found that over 25% of nonstudent participants chose an equal split in the dictator game. In our study, the ambiguous self-identification dimension complicated the clear-cut roles characteristic of standard dictator games. Therefore, although not every equal-split response can be interpreted as indicating a similar degree of identification with both continuers, the high percentage of such responses observed makes it likely that some participants truly did not prefer one continuer over the other. The "premium" in allocations to a future self versus similar others found by Molouki and Bartels [49] in the

same population of participants further speaks against a simple fairness explanation. Similarly, both monetary allocations and judgments of importance of the five criteria for identity and survival were distributed across the full range of the scale, which indicates an understanding of these attributes as gradual. However, the allocation of money did not have a real economic impact on participants and could be considered "cheap talk" [101]. Yet there are obvious ethical limitations in creating real economic incentives, and the distribution of allocations appears plausible and suggests that participants took the task seriously.

Further, giving participants the opportunity to split money between two continuers might communicate that it is appropriate to see both continuers as related to the original. Giving all one's money to one continuer (and thus endorsing that continuer as the sole legitimate continuer) might appear to be an (undesirable) extreme reaction to the question (see [102]). Our findings of transitivity violations might therefore be due to cueing as an artifact of our question technique.

Study 2 addresses these potential shortcomings by changing the central question from a continuous allocation task to an all-or-nothing decision. In addition, Attributes were ordered randomly, and the order of survival and identity questions was cross-balanced to test for order effects. Finally, we added questions on attributes beyond the five considered in Study 1, such as moral values, as these have been proposed as the primary indices of continuity (see, e.g., [65]).

## Study 2

In Study 2, we replaced the continuous task with a binary choice task, in order to assess the degree to which the results of Study 1 might have been shaped by our design choices [103]. All hypotheses formulated for Study 1 were also tested in Study 2, replacing references to monetary allocation by references to the new task.

### Methods

**Experimental setting and sample.**   Participants were again recruited via Amazon Mechanical Turk. The experiment was paired with a second, unrelated economic experiment in a Human Intelligence Task (HIT) lasting an expected 8–12 minutes, with a guaranteed payment of $1 and a possible bonus contingent on responses in the second task. The HIT was restricted to U.S. participants with matching browser location information and self-description and an approval rating of 95% or higher. Double participation was prevented using both the Unique Turker service and an analysis of IP addresses: In cases of multiple participation from one address, the first response was included if there was no temporal overlap with further attempts, resulting in a sample of $N = 821$ participants who were included in the analysis. One cluster of five suspicious responses was removed on the basis of near-identical IP addresses, identical browser signature, and near-identical comments. The mean age was 33.1 years ($SD = 10.0$ years) and 52.3% of participants were female.

The study was approved by the ethics committee of the MPI for Human Development in Berlin. Participants gave electronic consent after reading a consent form prior to beginning the study. The study was coded and conducted with Qualtrics, using JavaScript injection and including the libraries jQuery, jQuery UI, and Bootstrap to generate stimuli and tables.

**Experimental design.**   Study 2 also featured a fictional accident with a transporter pod (see section 1.2 in S1 Supporting Information for a full account). The description of the incident was nearly identical to the one in Study 1, with the same complete $2 \times 2 \times 2 \times 2 \times 2$ between-subjects factorial design, but the order of features was randomly determined for each

participant and the resulting distribution tables were created at run-time in the browser (see Fig 2).

Instead of being asked to determine insurance payments, participants in Study 2 were informed that "both persons will be compensated for the incident" and asked to make a decision about an inheritance payment:

*At the same time, the travel agency informs you that a distant relative of yours has died and that you have been chosen as the sole heir of the relative's estate. The inheritance cannot be split up into smaller parts and it cannot be sold or divided by other means.*

*Due to the circumstances of the incident, you are asked to decide which of the persons leaving the pod at your destination should be declared heir, person A or person B (it has to be either A or B; the other person cannot receive any part of the inheritance at any point in time).*

Participants were prompted to choose one of the two continuers. Rips and colleagues [17] argued that a closest continuer model should be able to predict the uncertainty associated with choosing, similar to models in decision theory based on weighted dimensions. The experienced uncertainty should be negatively correlated with the distance of continuers on the closeness metric. On a subsequent page, participants were therefore prompted to indicate the difficulty of this decision on a scale from 0 to 10.

Following this, participants responded to a series of statements regarding identity and survival after the hypothesized accident on a continuous slider scale (from "strongly disagree" to "strongly agree"). They also judged the importance of the five dimensions for survival and the importance of 14 partially novel dimensions for identity on a similar continuous scale (from "not at all important" to "extremely important"). The order of survival and identity questions was cross-balanced across participants.

Each participant answered the identity questions for each of the eight singular subdimensions in the five dimensions—body, appearance, psychology, personality, memory, knowledge, friends, and possessions—as well as for a random selection of six of 16 further dimensions, cross-balanced across participants. These 16 dimensions comprised five concerning relationships with others (family, loved ones, colleagues, general relationships with other people, and group membership), four concerning morality and values (moral values, virtues and vices, philosophy of life, religion), three concerning roles and profession (private roles, professional roles, and profession/job), two concerning demographics (gender and nationality), and two summary dimensions: mind and brain. These additional dimensions were chosen to explore to which degree the original dimensions might have led to an over- or underestimation of importance, e.g. if friends might be considered less relevant for personal continuity than close relatives.

Finally, each participant answered a series of questions measuring the tendency to reduce psychological to physical phenomena (see section 1.2.7. in S1 Supporting Information).

## Results

**Inheritance decision and difficulty ratings.** First, the binary decision between heirs was regressed onto the five factors constituted by the continuity or lack of continuity of the five dimensions. Table 1 shows the results of a binary logistic regression with the criterion "choice of person B" and each feature coded as 1 if present in person B and as 0 if present in person A (and therefore not in person B). Results showed a very strong effect of memory, a strong effect of psychology, weaker effects of body, and friends, but no significant effect of possessions.

**Table 1. Choice of person B in Study 2: Binary logistic regression, *N* = 821.**

| Variable | df | B | S.E. | Wald | p | e^B |
|---|---|---|---|---|---|---|
| Body in B | 1 | 0.45 | 0.17 | 7.44 | .006 | 1.58 |
| Psychology in B | 1 | 1.32 | 0.17 | 57.28 | <.001 | 3.73 |
| Memory in B | 1 | 2.25 | 0.18 | 165.77 | <.001 | 9.52 |
| Friends in B | 1 | 0.37 | 0.17 | 4.94 | .03 | 1.45 |
| Possessions in B | 1 | 0.17 | 0.17 | 1.01 | .32 | 1.18 |
| Constant | 1 | −2.39 | 0.23 | 105.97 | <.001 | 0.09 |

Fig 6 relates participants' choices to the location of memory and psychology for different splits of the attributes between the two continuers: The main effects of memory and psychology on inheritance choice do not seem to be dependent on the distribution of other features.

Combining the binary measure with the difficulty rating allowed us to compute an ANOVA comparable to the one in Study 1: A decision for person B was coded as 1, a decision for person A as −1, then multiplied by the inverse difficulty rating. The combined criterion has a middle point of zero, indicating maximum difficulty for either decision, a minimum value of −10 (decision for A with minimum difficulty) and a maximum value of 10 (decision for B with minimum difficulty). The results of a five-factorial ANOVA using the transformed values as

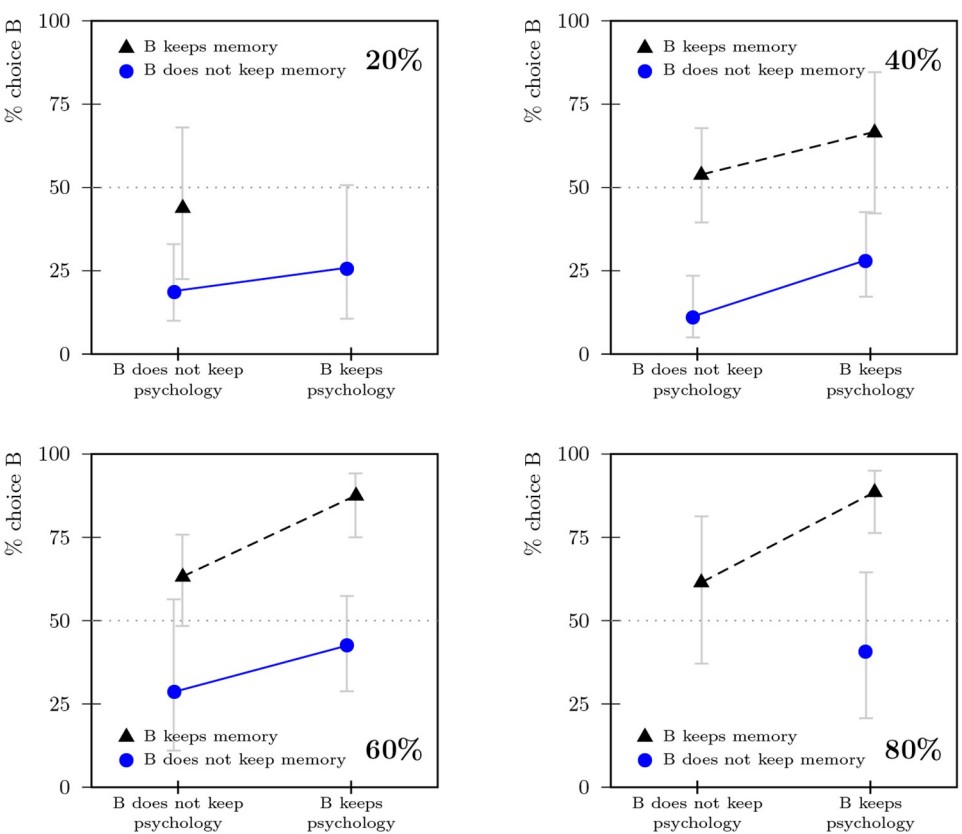

**Fig 6. Relative frequency (in percent) of choosing person B as heir.** Separate graphs summarize scenarios in which person B retains between one (20%) and four (80%) of the original dimensions. In each graph, allocations are shown for each possible division of the memory and psychology dimension. Note that B cannot retain both dimensions in the 20% condition and A cannot retain both dimensions in the 80% condition. Whiskers show the 99% confidence intervals around the means.

criterion were comparable to those of the logistic regression: We found strong main effects of memory ($F$(1, 789) = 295.38, $p < .001$, partial $\eta^2 = .27$) and psychology ($F$(1, 789) = 97.07, $p < .001$, partial $\eta^2 = .11$), weaker main effects of body ($F$(1, 789) = 11.99, $p < .001$, partial $\eta^2 = .02$) and friends ($F$(1, 789) = 7.63, $p = .006$, partial $\eta^2 = .01$), and no main effect of possessions ($F$(1, 789) = 1.62, $p = .20$, partial $\eta^2 = .00$). There were no significant interactions (see Table I in S1 Supporting Information).

The inheritance decision data thus again provide strong support for Hypothesis 1b (psychology) and Hypothesis 1c (memory). There is weaker support for Hypothesis 1a (body) and Hypothesis 1d (friends) and no support for Hypothesis 1e (possessions). Shifting the focus to which combination of factors makes decisions more difficult, we ran a second ANOVA with decision difficulty as criterion (see Table J in S1 Supporting Information). The only substantial effect was an interaction between location of memory and psychology ($F$(1, 789) = 56.31, $p < .001$, partial $\eta^2 = .07$; see also section 2.2.2. in S1 Supporting Information), followed by a weaker interaction between location of body and memory ($F$(1, 789) = 6.9, $p = .009$, partial $\eta^2 = .01$).

**Identification with continuers.** Parallel to the analysis conducted in Study 1, we calculated two ANOVAs with the five location variables as factors and the two statements describing identification with person A or person B as criterion (see Table K and Table L in S1 Supporting Information). The strongest effect in both ANOVAs was again the main effect of memory ($F$(1, 789) = 133.13, $p < .001$, partial $\eta^2 = .14$ for person A and $F$(1, 789) = 125.57, $p < .001$, partial $\eta^2 = .14$ for person B), followed by the main effect of psychology ($F$(1, 789) = 78.89, $p < .001$, partial $\eta^2 = .09$ and $F$(1, 789) = 69.81, $p < .001$, partial $\eta^2 = .08$). In addition, a significant, but weaker, main effect emerged for friends in both ANOVAs ($F$(1, 789) = 4.07, $p = .04$, partial $\eta^2 = .01$ and $F$(1, 789) = 4.23, $p = .04$, partial $\eta^2 = .01$), and a significant main effect emerged for body for person A only ($F$(1, 789) = 5.76, $p = .03$, partial $\eta^2 = .01$ and $F$(1, 789) = 3.54, $p = .06$, partial $\eta^2 = .00$). No interaction effect was consistently significant for both ANOVAs: There was one significant two-factorial and one three-factorial interaction for each. In summary, these results mirror the results of the preceding analysis of the inheritance decision.

**Explicit weighting of dimensions.** Fig 7 shows participants' ratings of the importance of the full set of 24 possible dimensions for determining identity in order of average assigned importance. Together with the pattern of intercorrelations (see Table O and Table P in S1 Supporting Information), this approach allows us to compare the attributes featured in our scenarios with alternatives and to evaluate the relationship between jointly presented features. Memories and personality (here separated from knowledge and psychology) are considered the most important features, closely followed by knowledge and psychology.

The separate evaluation makes it possible to compare the ratings for otherwise joint dimensions (see section 2.2.3. in S1 Supporting Information): Appearance and body have similar mean ratings and are highly correlated ($r = .76$, $p < .001$). Similarly, personality and psychology ratings are correlated ($r = .53$, $p < .001$) and, to a lesser degree, memory and knowledge ratings ($r = .37$, $p < .001$), and personality and knowledge ratings ($r = .37$, $p < .001$). While questions differed from Study 1, we conducted a cluster analysis on the weightings of the eight separate dimensions (see section 2.2.7. in S1 Supporting Information). Most clusters could be replicated in the new cluster analysis. The subdimensions combined in the scenario were rated very similarly in all but one cluster.

Moral values, philosophy of life, and virtues and vices were considered less important than the dimensions described above, but more important than the rest. These findings are inconsistent with previous studies [65] and are considered in more detail in the general discussion. There was a substantial gap between this group of dimensions and religion, indicating that

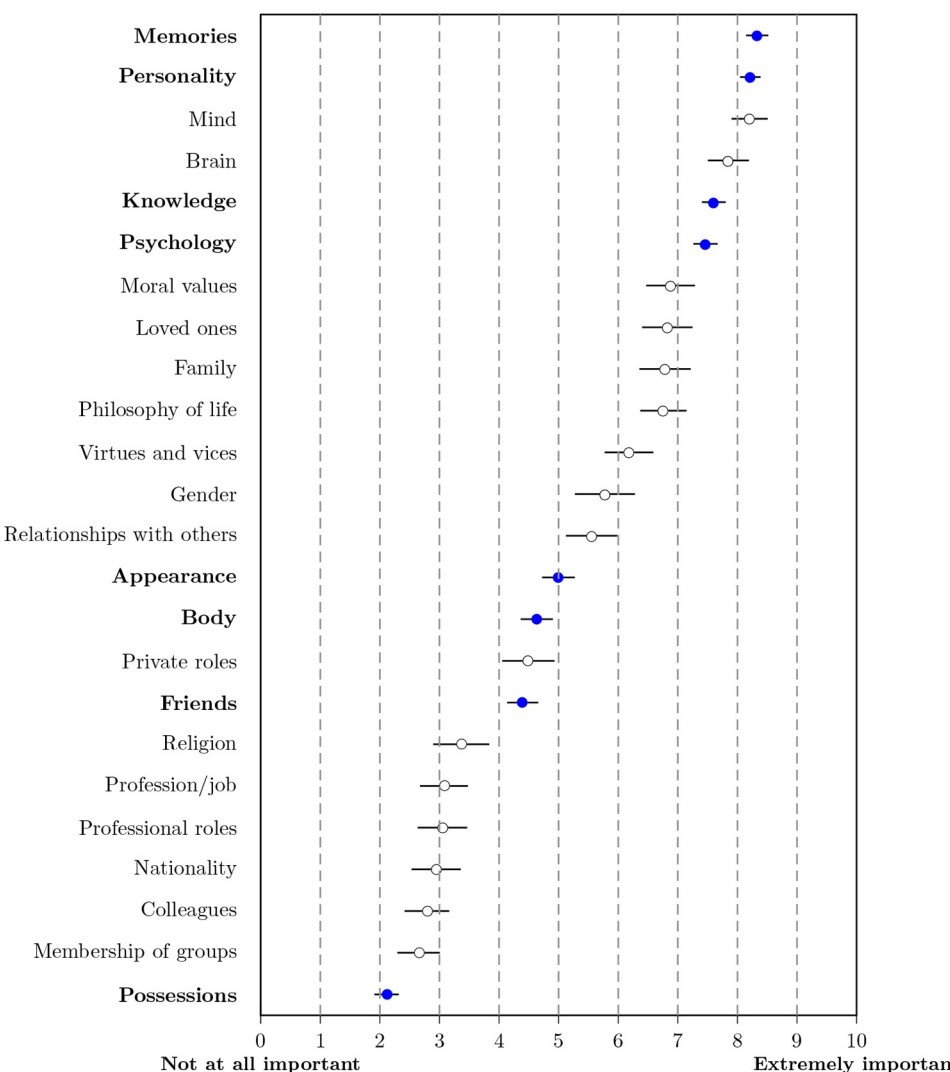

**Fig 7. Importance assigned to attributes for determining identity between persons ($300 \leq N \leq 817$) in Study 2.** Dots and bars show means and corresponding 99% CIs. All participants rated the eight dimensions marked with filled dots and a randomly selected subset of six other dimensions (balanced across participants). Elements of the five scenario dimensions are labeled in bold.

participants did not consider moral values and virtues to be strongly dependent on religion. In addition, ratings for moral values, philosophy of life, and virtues and vices correlated highly with those for personality, psychology, and knowledge and, to a lesser degree, with those for memory and friends, whereas religion ratings were substantially correlated with friends and possessions ratings only.

With respect to attributes describing relationships with other people, family was regarded as most important, followed by relationships with others in general and friends, and—much less important—colleagues. Ratings for family were at least moderately positively correlated with those for all features used in the scenario except psychology and personality. Low importance ratings were given for all attributes relating to profession, as well as for nationality and possessions. In fact, possessions were rated least important, echoing the low importance of this factor in determining decisions in both scenarios.

Participants' responses to the survival questions were generally in line with those reported in Study 1 with minor deviations. Order effects for survival and identity questions were small and did not come close to change the interpretation of results (see section 2.2.5. in S1 Supporting Information).

**Personal reidentification after fission.** Comparing participants' ratings of the post-questionnaire items between studies (see also section 2.2.4. and Table Q in S1 Supporting Information), the largest differences between the studies were only about half a scale point in size, yet there was a clear pattern to their direction. In Study 2, the maximum identification with one of the two continuers was higher ($t(1515) = 7.82$, $p < .001$, two-tailed test), the minimum identification was lower ($t(1515) = -4.00$, $p < .001$), endorsements of both personal survival ($t(1514) = 6.73$, $p < .001$) and being the same ($t(1510) = 4.55$, $p < .001$) were stronger, and there was less agreement with the statement that the original person does not exist after the incident ($t(1509) = -3.03$, $p = .002$). There was no significant difference in endorsement of the statement that the original person was now two persons ($t(1514) = -1.02$, $p = .31$). This pattern of results is consistent with the idea that the task of splitting money led to a decrease in identification with one continuer (and not the other), but it does not completely explain the phenomenon of intransitivity: Most participants responded with an average minimum score $>1$ to questions regarding identification with continuers and of $>2$ to the question regarding being two people after the incident. In other words, some participants again committed a violation of transitivity in identifying the original person with more than one continuer. The continuous scale used in Study 1 is not enough to explain this.

Table 2 summarizes means and intercorrelations of the items tapping participants' perspectives on identity relations after fission. Answers to questions regarding survival and nonexistence were highly negatively correlated ($r = -.57$, $p < .001$), and the acceptance of being two people was positively correlated with the minimum identification with either person ($r = .35$, $p < .001$).

An ANOVA with agreement that the original person was now two persons as the criterion (see Table M in S1 Supporting Information) again showed a significant interaction of memory and psychology ($F(1, 789) = 10.48$, $p = .001$, partial $\eta^2 = .01$), but also interactions between memory and friends ($F(1, 789) = 15.60$, $p < .001$, partial $\eta^2 = .02$), body and memory ($F(1, 789) = 10.00$, $p = .002$, partial $\eta^2 = .01$), psychology and friends ($F(1, 789) = 5.38$, $p = .02$, partial $\eta^2 = .01$), and a four-factorial interaction of body, psychology, friends, and possessions ($F(1, 789) = 6.07$, $p = .01$, partial $\eta^2 = .01$). An ANOVA with agreement to nonexistence as criterion

**Table 2. Correlations, means, and standard deviations of the six post-questionnaire items and decision difficulty scale, Study 2.**

| Item | Mean | SD | 1 | 2 | 3 | 4 | 5 | 6 |
|---|---|---|---|---|---|---|---|---|
| 1. You have survived the incident | 3.42 | 1.46 | — | | | | | |
| 2. You are the same person as before the incident. | 2.02 | 1.62 | .39*** | — | | | | |
| 3. You are person A/B after the incident. (Max) | 3.52 | 1.23 | .33*** | .04 | — | | | |
| 4. You are person A/B after the incident. (Min) | 1.18 | 1.04 | −.11** | −.10** | −.06 | — | | |
| 5. You do not exist after the incident. | 1.73 | 1.55 | −.57*** | −.36*** | −.21*** | .21*** | — | |
| 6. You are two people after the incident. | 2.19 | 1.60 | −.15*** | −.27*** | −.06 | .35*** | .26*** | — |
| 7. Difficulty of decision (1–10) | 4.62 | 2.83 | −.13*** | −.14*** | −.12*** | .32*** | .22*** | .19*** |

$N = 821$,

\*\*\* $p < .001$,

\*\* $p < .01$,

\* $p < .05$.

(see Table N in S1 Supporting Information) similarly shows an interaction of memory and psychology as strongest effect ($F(1, 789) = 22.34$, $p < .001$, partial $\eta^2 = .03$).

Inspecting correlations with the difficulty scale gives further insight into these results: Variation in the perceived difficulty of the decision was negatively correlated with endorsement of survival and staying the same after the incident and positively correlated with a higher minimum identification with both, as well as with endorsement of being either both persons or none after the incident (see Table 2). Consistent with the predictions of the closest continuer theory, ratings for identification were positively related to the percentage of traits retained, and for scenarios with 60%:40% splits of dimensions, ratings for survival and for being the same were the lowest and ratings for being two people and for being nonexistent were the highest (see section 2.2.9. in S1 Supporting Information for visualizations and tests).

Given the presentation of the same six questions as in Study 1, we attempted to replicate the cluster analysis conducted before (see section 2.2.8. in S1 Supporting Information). The new cluster analysis shows a reasonable match for all but one original cluster. Again, cluster membership stood in a significant relationship with features of the encountered scenario.

## Discussion

Changing the task to an all-or-nothing decision allowed us to differentiate more clearly between features: Memory was shown to influence inheritance decisions more strongly than psychology, and the importance of both factors for determining identity was confirmed. When fairness considerations were ruled out, we found some evidence for the importance of both body and friends but, again, no evidence for an effect of possessions. Thus, our results provided strongest support for Hypothesis 1c (memory), strong support for Hypothesis 1b (psychology), limited support for Hypothesis 1a (body) and Hypothesis 1d (friends), and no support for Hypothesis 1e (possessions). Again, there was support for Hypothesis 2 and Hypothesis 3.

Although the level of agreement with statements violating transitivity was lower than in Study 1, participants again endorsed these positions. Endorsement of being both persons after the incident was positively related to perceived decision difficulty, which was in turn strongly related to the separation of original memories and psychology.

## General discussion and conclusion

In both of our studies, a distinct pattern emerged in the order in which participants rated dimensions commonly associated with closeness and identity. Although fairness considerations moderated the degree to which money was split between the two continuers in Study 1, clear main effects nevertheless emerged for two of the five dimensions: personality/psychology and memory/knowledge. Answers to survey questions further revealed that most participants espoused a psychological theory of identity. Many participants did not use personal identity in the way that philosophers would consider normative: They endorsed the idea of being identical with two distinct continuers at the same time. This tendency was most pronounced when personality and memory were split between the continuers. Study 2 provided additional, yet weaker, support for the importance of body and social relations in the context of inheritance claims, which was also found in a closer investigation of individual response patterns. Contrary to the extended self hypothesis [13, 87], both studies found little evidence for possessions playing an important role in laypeople's theories of personal identity and closeness. Qualitative responses and post-questionnaire responses indicated that the closest continuer theory is a productive framework for analyzing personal reidentification and subjective theories of identity. Although our experimental paradigm cannot address all

conflicts and ambiguities surrounding the difficult construct of personal identity, we consider it a promising approach for studying subjective closeness metrics and intuitions about identity.

Given the interdisciplinary nature of our study, we discuss both psychological and philosophical contributions and implications of our work, as well as possible objections from both sides of the aisle. We start by discussing specific contributions to the discourse on identity within psychology, then address some strengths and weaknesses of our scenario, and conclude by illustrating the general relevance of research on personal reidentification.

## The crisis of identity in the social sciences

Outside philosophy, "personal identity" can be seen as a "notoriously haphazard concept" (in [104], p. 521). Numerous mutually incompatible meanings have been attached to personal identity across disciplinary discourses over the past 60 years, resulting in a high degree of ambiguity and a lack of conceptual clarity. Hitlin [105] even called the term "untheorized in much social science work" (p. 120). One could argue that large parts of the discourse in psychology (and related disciplines) equate the entity to which philosophers ascribe personal identity *with* personal identity. In other words, when referring to personal identity, they describe the person or the self (sometimes bundles of roles and categorizations) or—to compound the difficulties—self-perceptions or perceptions of others regarding this entity, sometimes simultaneously. Heiphetz and colleagues [106] assumed that most psychologists associate identity with individuality or group membership (p. 744). A theory of personal identity must untangle these elements in order to understand what entity is considered to continue. Attempts have been varied. Erikson [107] somewhat circularly integrated the sense of one's identity into the concept of ego identity. For [108], answers to the question "Who am I?" would directly relate to personal identity, for [109] personal identity was a self-construal that differentiates a person from others, for Hitlin [105] it "refers to the various meanings attached to oneself by self and others" (p. 120), and for Chen and colleagues [53] it "provides people with norms to follow, scripts for behaviors, and ways to interpret their actions [. . .] and affects a wide range of in-lab and real-world decisions" (p. 1398).

There are many more examples of divergent and in many cases, mutually incompatible, use. Naturally, "attesting sameness of a person depends upon what 'person' is taken to mean" (in [21], p. 744), but "identity" in the sense of sameness should not be confused with "person." In short, while all of the many approaches might tackle sensible and important research questions, there is reason to doubt the usefulness of moving away from the term's original meaning, which continues in philosophy: The medieval Latin term *identitas* meant sameness and lack of variation. "Identity" was still mostly used in this sense by James [13], who described the sense of personal identity as the "consciousness of personal sameness" (p. 217) and regarded this sameness as the same relation as for objects: "The intellectual operations seem essentially alike, whether I say 'I am the same,' or whether I say 'the pen is the same, as yesterday' "(p. 218). Chandler and colleagues [61] considered identity in this sense "among the oldest of our ideas" (p. 6). All the incompatible uses of "identity" listed above could well be replaced by more precise terms.

While philosophers might disagree on the right theory of personal identity and the right concept of persons, they show much less disagreement regarding the identity relation. Strohminger and Nichols [65] ascribe the idea that "identity can be graded and relative" to Parfit and Hume (p.159, footnote 1). It is correct that both authors critically examine idea of the identity (in the sense of sameness) of a person from birth to death. Hume [4] in his *Treatise* famously retracts his attempt to analyze personal identity in the appendix [110], as given his

phenomenalistic methodology [111] nothing in a person over time qualifies to stay exactly the same, it is *because* identity cannot be graded or relative that he runs into this problem.

Parfit [6] would consider specific events to be identity-disrupting, so that "[w]e may regard some events within a person's life as, in certain ways, like birth or death" (p.328), but this changes the set of entities for which identity holds, not the relation. Indeed, the central dilemma posed in [6] could be easily solved if identity were allowed to be "graded and relative".

Our participants' positions on personal identity might have been influenced by the discourses in philosophy or psychology, but it could even be called into question whether they had formed explicitly defined beliefs about the self at all [112]. The scenarios we applied used practical problems (tied to the targeted meaning) to test theories of identity without actually featuring the word "identity," thus sidestepping any such ambiguity and confusion. We later ask a question concerning the identity *between* people, which is incompatible with non-relational understandings of identity. By asking participants to make decisions between continuers, we were able to focus our analysis on numerical identity without restricting or contextualizing personal identity further or assuming familiarity with a specific understanding of identity. This approach might imply that our participants' answers do not directly inform any of the specialized discourses within philosophy or psychology; nevertheless, those answers are important in understanding how participants conceptualize personal identity, closeness, and the self. Any appeal to generally shared intuitions needs to be checked against the range of intuitions expressed by the general public. For this reason, our participants were recruited from Amazon Mechanical Turk. Although the sample was restricted to one nationality, our participants showed more sociodemographic variance than student convenience samples [113], and we were able to overcome the problem of small samples [114, 115].

## Separating closeness from identity

The closest continuer theory offers the advantage of naturally sorting judgments of closeness and judgments of identity into one decision procedure, while keeping these judgments conceptually separate. This framework is therefore a good basis for organizing the multitude of research approaches to personal identity in the past and likely future. We argue that the majority of approaches have focused either on the closeness metric alone or on the question of the minimum closeness threshold for accepting a continuer (Case 2 vs. Case 4 in the decision heuristic).

Strohminger and Nichols [116] asked whether a person was recognized as the same person given psychological changes, whereas [106] measured the perceived change in a person on a scale from 0% (same person as before) to 100% (completely different). In many other studies endeavoring to target identity the central questions focused likewise on similarity to the original or degrees of change [21, 65, 117, 118], similar to typical questions in the "future self" approach, that elicit the degree of overlap between present and future self (e.g., [45, 47]). These questions mostly correspond to measures of closeness, which are an important element of a theory of personal identity but cannot be equated to personal identity. Starmans and Bloom [5] take the more extreme position that questions asked in [65] are only about similarity or qualitative identity (see also [15, 16]) and not personal identity, and that no degree of dissimilarity through personal change could disrupt personal identity. Parfit [6] acknowledges that certain life events might count as birth and death of a person: A complete disruption of psychological connectedness would also disrupt personal continuity. For Nozick [14] this situation would correspond to Case 2.

Defining the features of the closeness relation might also help to organize findings about identity explained by forms of essentialism. Essentialism, the belief in some mostly unspecified core or unobservable substance conveying identity [61, 119], has been invoked to explain why children and adults believe that objects and animals remain stable when their properties change [120], or why objects are judged to stay the same after a complete replacement of their parts [26]. The analysis of closeness relations might help to better understand these findings, as well as define which parts of persons are considered "essential," the conditions under which present and future selves are considered to be overlapping, and how personal change is assessed.

Most empirical studies investigating changes to the self and personal identity utilize scenarios that restrict the number of possible continuers to one. These scenarios tend to correspond well to real-world scenarios and offer some insight into subjective closeness metrics, but allow the exploration of personal identity only to a limited degree. We would like to focus on two concerns inherent in intertemporal change scenarios. First, participants engage with scenarios featuring gradual change with expectations regarding ordinary change processes. These expectations may become more important than judgments of closeness. Some expectations are certainly reasonable: For example, most observers would identify an infant as the same person as the resulting adult, even if the two do not share many properties, since these changes are gradual and expected. Changes that would break core assumptions about aging—such as a younger person being the continuer of an older person—would rightfully lead to a rejection of identity. Scenarios featuring change are always checked against the array of expected gains and losses, and these expectations are likely to reflect optimistic biases [117, 121–125].

More importantly, questions about a single continuer are limited in that they can explore the minimum threshold for accepting a continuer, but do not allow for differentiation between continuers that jointly reach or miss this threshold. A theory of personal identity needs to be able to address both problems.

## Intransitivity and alternative concepts of persons

In both studies, participants were told to imagine that they were about to be split into two continuers [17, 52] who would each inherit different features or attributes from the original person, similar to [53] (Exp. 3 and Exp. 4). The choice between continuers in a multifactor approach [126] allowed us to compare the strength of competing claims that might be independently sufficient for warranting continuity [26]. This setup also allowed us to study otherwise inaccessible aspects of subjective theories of identity. We were able to connect the judgments of many participants to the predictions of the closest continuer theory. We also presented evidence that intransitive judgments show interindividual variability and further depend on the relative strength of competing claims, which to our knowledge had not previously been demonstrated for personal identity judgments. The scenario structure is modular enough to allow for the inclusion of further dimensions and the testing of further conditions, which may help to gain insight into theories and the decision process in the future.

Some participants might have nonstandard perspectives on the metaphysical nature of objects and persons [33]. If two continuers can be considered to be the same as the original without a sense of contradiction, it is not necessarily the understanding of identity that is responsible for this conflict. It is possible—albeit perhaps unlikely—that participants instead endorse alternative views of persons and conceptualize them as time worms, life histories, or branching persons [127, 128]; or allow for two persons cohabiting one body before fission [58]. One might criticize our questions for leading participants to a nonstandard perspective of persons by using continuous scales for eliciting opinions about states after fission that

cannot be true in degrees. But we hold that agreement to statements in surveys is not equivalent to asserting their truth value. Many survey questions measured on Likert scales that participants on the platform face on a daily basis only allow for binary answers under a strict interpretation. Surveys responses rather reflect the degree of a participant's certainty in the answer and conviction. As [34] concluded, "although it is *false* to say so, it might *not* be *wildly* false to say, for example, that [. . .] I survive while scattered in two main parts" (p. 258). We consider it unlikely that our questions lead participants to change their understanding of the terms in our study. Very similar results to ours have been obtained with a multiple choice format by Weaver and Turri [52], which casts further doubt on the response-format objection (see also [17]).

Our study might not convince a revisionary metaphysicist who believes that laypeople are fundamentally mistaken in their beliefs about personal identity, and they will likewise be critically received by philosophers who value consistency and logic irrespective of consensus in adjudicating claims. We did not confront participants with any of the paradoxes that have been created to attack the consistency of positions they endorsed in their own answers, and for their part, participants did not seem to differentiate between criteria for identity and criteria for survival: Their average ratings of the importance of the various dimensions were similar for both. The scenario with two equally close continuers differs from that with no continuers, as the former might be able to realize at least some of the goals people care about in survival. They could continue and finish projects [6] and they could extend "the causal influence of our psychology into the future *at least once*" (in [51], p. 309). Participants might have chosen to equate this sense of continuity with actually being those two persons in their answers.

## The fission scenario as scientific fiction: Benefits and validity concerns

Not all philosophers embrace the use of hypothetical scenarios in studying identity [129–131]. Some are critical of fission thought experiments, arguing that the intuitions derived from such fantasy accounts—in which anything goes—violate natural worlds and are therefore not valid measures of how people conceptualize identity in natural settings (e.g., [132]). Like contaminated test tubes in biochemical experiments, results obtained with faulty scenarios would have to be considered dubious [131]. Scholl [130] further argued that reactions to "bizarre scenarios" with forced responses might "tell us more about heuristic getting-through-the-experiment strategies than about actual metaphysical intuitions" (p. 580). One could compare these scenarios to visual illusions that generate experiences that are at odds with reality—but these experiences nevertheless often provide insight into the mechanisms that generate the illusion. For example, size and distance illusions reveal the computation the brain uses to calculate physical dimensions even in normal cases. Likewise, intuitions derived from cognitive processes during these unreal and outlandish examples may not be necessarily meaningful when measured against the constraints of reality [130] but rather cast light on how we use and reason with the concept of identity. While the participants' open answers showed a degree of confusion in a minority of subjects, most answers reflected a well-considered and principled approach to the questions. Studies in experimental philosophy often feature unusual scenarios, and replicability in this sub-field compares favorably with replication rates in other domains of psychology [133].

Our science fiction scenario might be criticized for its lack of realism. It makes crucial assumptions that contradict the scientific understanding of how the components that we neatly separated in the story interact and co-depend on each other. According to Harle [68], a brain cannot be considered to be independent of the body; it would immediately adapt to a new body, rework the inner representation, and react to changes in social responses. Brain and

body are interdependent in complex ways [39, 51, 63]. Motor learning [134] also involves two components, body and memory; muscles cannot work without neuronal input. The body and appearance component used in our studies could be understood to either include or not include brain matter. If included, all other components would depend on this component, as there can be no psychology without consciousness. If, alternatively, higher mental functions and consciousness are regarded as part of psychology or memory (which seems to be the approach taken by at least some participants), the brain, as a bodily organ that makes psychological functioning possible, can still be seen as source of tension in the scenario framework.

Participants responded to our scenario only from a first-person perspective. First-person evaluations of identity and survival might differ from third-person evaluations [128, 135]. For one thing, many legal, practical, and social concerns can be fulfilled by a person who is a spontaneous true copy of the original person. How much participants care about the disruption of continuity might therefore differ depending on whether the replaced person is *them* or a neutral other person [30]. Rorty [18] distinguished between an external observer's perspective on individual identification and an individual's internal perspective; features essential to an individual's self-perspective might be irrelevant for an observer, and not all philosophers assume that first-person judgments have final authority [136, 137]. On the other hand, at least one study explicitly testing for the effect of perspective on intuitions about identity based on [28] found no substantial differences between a first-person and a third-person perspective [31]. At the same time, continuers were introduced from a third-person perspective. This shift was necessary to avoid a pre-judgment of the question how the original person relates to them, but might create a perceived distance. This could make it easier to give a negative answer to the survival question, but would have a symmetric influence on identity responses for both continuers.

It is less clear which perspective is better suited to evaluate claims about identity and survival. We assumed that involving the participant would be the best way to increase attention to the situation and diligence in responding. We induced a connection between our categories and their real-world instances, as perceived by the participants. A participant evaluating the importance of "body" in the money allocation problem will thus take the perception of her own body into consideration, which may lead to different results than when considering the importance of *a* body. This fact might play into our finding that a subgroup of participants placed a negative value on the continuity of the body.

On the other hand, our scenario avoids a complication common to many other fission scenarios: By splitting possessions and friends between survivors, it sidesteps the problem of non-sharable singular goods [34] with symmetric claims from two sides. These claims include access to special objects and, more significantly, the chance to engage in special relationships with others [51]. On the flipside, as Schechtman [138] argued, the scope and time frame of fission scenarios does not allow for societal reactions to the products of fission, which might include changes to the concept of persons and identity. Our approach avoids the confound of money being allocated for reasons of loss or pity, as both survivors emerge with their lives, bodies, and environments fully intact (if not unchanged). Williams [28] predicted a framing effect in the understanding of the scenario, a type of "leading the witness," with a reversal of intuitions regarding identity depending on whether the story is told as involving body-swapping (transferring a person's memory to a new body) or mind-manipulation (creating new memories in a person's body). Empirical work has confirmed these predictions [31]. To counteract such potential framing effects, we used parallel language for all components varied in continuers; none of the components was privileged in the description of the scenario. Further, similar tensions involved in our scenario are discussed in section 1.3. in S1 Supporting Information.

Invoking hyperspace travel buys some degree of freedom, but at the cost of physical implausibility. However, factual impossibility does not prevent imaginability, and as Johnston [139] argued, "such *per impossible* thought experiments might nonetheless teach us about the relative importance of things that invariably go together" (p. 601). It is precisely because some aspects are not easily separated in realistic scenarios that we chose a fantasy scenario, allowing us to explore intuitions whose tests would otherwise be confounded.

Our scenario remains within the conventions of popular films (e.g., *Total Recall* or *Blade Runner*—both of which fittingly now exist in two versions) that deal with cases of copied or artificial memories and identities [140]. Body and mind transfer were considered intelligible by Locke [9], and the nature of personal identity is a recurrent theme in literature. Many readers have appreciated Franz Kafka's tale of Gregor Samsa's sudden metamorphosis into an insect [12] or Ovid's metamorphosed subjects who survive the transformation [39]. Children become acquainted with bodily transfer in fairy tales like *The Frog Prince* [23, 120] or Hans Christian Andersen's *The Little Mermaid* [39]. Johnson [141] takes this mere imaginability as an argument against declaring bodily continuity as a logical precondition for personal identity. Our scenario is no more fantastic than other thought experiments that have been employed to disentangle identity from its natural correlates—through neurosurgeons [6, 38, 51, 128, 142], amoeba-like duplication [127, 143], cloning [144], parallel universes [30], or even swamp-beings [66]. To come full circle, some of the philosophical conceptions and puzzle cases are reproduced in cultural creations and thereby further embedded into cultural consciousness [39]. It is possible that there is no theory of personal identity that would be able to "satisfy all intuitions about all devisable scenarios" (in [31], p. 297), but the advantage of imaginary scenarios lies in their power to isolate phenomena, which makes it possible to attend to specific aspects of our concepts [145]. In our scenarios we can separate changes from their ordinary causes and study decisions that may not occur in the world, but that probe concepts we apply to the world. In addition, our use of a novel paradigm limits the danger of previous exposure to similar questions and potential confusion, which is a concern with crowdsourced participants [146]. We therefore maintain that there is value in our approach of pitting dimensions that are generally accepted as dimensions of closeness against one another.

## Dimensions of closeness

Our scenario bundles features into dimensions that might be further differentiated. For example, in contrast to [65], we did not differentiate between personality and psychology, on the one hand, and moral values, on the other. Evidence from several studies considering real-world personal transformations has indicated that identity judgments are most heavily influenced by changes or non-changes in moral values [65]. Changes in morality were judged to be more relevant than changes in (non-moral) personality attributes or memory. In a similar vein, Strohminger and Nichols [116] found that changes in morality in patients with neurodegenerative diseases strongly determined changes in perceived identity. Nunner-Winkler [21] reported on a study asking participants which changes would lead them to see themselves as a different person. Ideas about right and wrong and sex membership were considered to be quite important; appearance and money were considered less relevant (although some participants rated looks to be important, consistent with our distributional results).

The distinction between moral and nonmoral traits is somewhat ambiguous (e.g., conscientiousness was considered as a moral trait rather than a personality factor in [65]). One person's morals do not and cannot exist in a social vacuum, moral consensus is central for co-ordination, affiliation and conflict resolution. Morality stands in complex relations to beliefs, values, behaviors and communities. It also depends on memory in nontrivial ways. Some of the

induced changes in the scenarios even involved the loss of the moral faculty with a likely ripple effect reaching other dimensions of the self. If this perspective is true, the relevance of morality for personal identity might lie in these possibly disruptive consequences of changing one's morals in relation to one's environment and not because of its self-defining importance. Evidence for this interpretation is found in two studies demonstrating that changes in widely shared (and therefore less unique to the individual) moral values are considered to lead to more changes to the person than changes in controversial moral beliefs [106, 147]. For controversial moral beliefs, that might be considered most defining and informative for describing a person's self, the effect was weaker than for memory. Also, the changes in memory induced by our scenarios would induce both errors of omission and errors of commission, which can have differential impacts on moral behavior [148]. In contrast, some studies focus mostly on omission errors due to memory changes (e.g., [118]), which are describes as having more limited effects on behavior towards others than changes in morality. In our scenarios, dimensions are replaced by random sampling from the participant's reference population, which is a different operationalization of change. Heiphetz and colleagues [147] showed, for example, how the perceived change was mediated by perceived disruptions of friendships.

Some argue that morality is not even conceivable *without* personal identity [6, 10, 58]. Most people also seem to have inflated beliefs of their own morality [149]. In separate evaluations, our participants ranked memories and psychology to be more important for identity than moral values. Nonetheless, a further decomposition of the broad headings we used in our study would be feasible and interesting in future research. In particular, the role of moral traits and behavioral tendencies could be considered separately, even within a similar factorial setup as the one we employed.

The social dimension could be further differentiated, as well. Parents were considered to be more important than friends in Study 2, and Nunner-Winkler [21] reported similar findings. Of course, parents influence a person directly through the transmission of genes as well as indirectly through instruction and parenting behavior; changing one's parents cannot be considered a merely social manipulation and could well have an impact on every other dimension.

In our scenario, changes in memory are considered universal and all-or-nothing. In real life, however, memories of self or self-knowledge seem to be better preserved than other knowledge, even in semantic dementia [20], and a subjective belief of self-persistence is demonstrated by patients with Alzheimer's disease [150]. Alternatively, the sense of self may be impaired while episodic memories stay intact—as in the case of R.B. [67]. Further, separating specific psychological aspects or memory from a person's social context and network of activities might prove impossible in practice [18]. There is also some overlap between criteria based on psychology and memory, but under the assumption that two organisms with the same memories might nonetheless differ in personality and psychology (e.g., based on differences in needs, intentions, values, or goals), it is not necessary that the criteria coincide. In fact, the psychological continuity criterion has been proposed as a critique of a narrow Lockean focus on memory [11].

Critics of our scenario might further object that our random collage of features in the two continuers destroys the causal connection between past and present states necessary for identity [6, 30]. Preschool children already individuate objects and persons spatio-temporally [23, 151] and, following Sagi and Rips [152], causal histories receive special attention in linguistic disambiguation in discourse. In all our scenarios (except the two extreme cases with exact duplicates), change in characteristics was induced by an accident, an unusual life event that disrupts spatio-temporal continuity. This fact might strengthen impressions that identity is not preserved. According to data reported in [21], for example, participants regarded changes in attitudes or beliefs that were due to normal life experiences as non-consequential for

identity judgments—as opposed to changes induced by brainwashing, severe medical conditions, or accidents. Therefore, the nature of the transformation might play a role in our participants' judgments. Note that both continuers underwent the same procedure, so this factor cannot explain differential assessments. Although the abruptness and symmetry of the original person's transformation prevents the application of spatio-temporal continuation criteria, participants might still construct "fictive causal histories" [153] to assess which of the two continuers might have the better chance of being the result of changes within an ordinary life.

Finally, for a continuer to acquire a random set of possessions, these would have to materialize from somewhere. Our scenario also assumes that this change in possessions can leave memory, psychology, friends, and appearance untouched. This is incompatible with the reality that some of our memories are intertwined with objects in our possession and the difference between owning or not owning status symbols, for example, can impact self-value, build and burn bridges with others, and change perceptions of their owner.

### Towards a process model of re-identification

Our studies allow to make some progress in the analysis of decisions involved in determining personal identity. Like Rips and colleagues [17], who develop the causal continuer model based on Nozick's theory, we are interested in the decision process. Decision processes, as implemented by human beings, are often insufficiently described by functions merely predicting decision outcomes. A further analysis of the decision processes needs to address questions of information search: Which persons are considered as continuers? When and why is the search for possible continuers stopped? Which dimensions are considered in the subjective closeness metric, and how are these dimensions integrated? We showed some results compatible with decision-making following the closest continuer logic. Is there further evidence for the three steps being followed in a specific sequence—the fast-and-frugal tree in Fig 1 would not yield different outcomes if the first three levels changed their relative position—and how stable is this process across individuals? A structurally similar model of decision making has been proposed for explaining the phenomenon of choice deferral [154, 155]. When faced with a selection of possible alternatives, choice in the 2S2T-model [155] is deferred for one of two reasons. First, none of the options is good enough and surpasses a decision threshold or second, too many options are good enough, surpassing the threshold but it becomes difficult to choose the best option. Of course, personal re-identification is not simply preferential choice but the analysis of the decision process might still be informed by the analysis of related or parallel processes in other domains.

While the mathematical form of weighted-additive linear models implies weighting and adding, many other operations, such as lexicographic stepwise procedures that ignore (sometimes most of the) variables in the equation [35] would still be captured by this model [156]. Brook [12] argued for a model of personal re-identification that starts with psychological factors and only considered other dimensions if the information is missing (or inconclusive). Variance in choosing and applying criteria might again be related to other individual differences [41, 42]. Based on the variations in our chosen design for this study, it is not yet possible to build cognitive models of participants' decisions. It is, for example, unclear whether an appropriate model should be stochastic, as in [17], or deterministic.

Our scenarios varied factors that should mostly influence the assessment of closeness and only indirectly the decision-making based on these assessments. Future research could shift this focus to the subsequent stages of the procedure. Thus, specific exit nodes of the decision tree in Fig 1 could be investigated. For example, is there a minimum level of closeness required for participants to determine that any of the continuers is identical to the original person? Do

participants share the intuition that a fission resulting in multiple exact copies does not preserve identity, and would this depend on the level of closeness? What type of difference is considered to be sufficient to single out a closest continuer?

Both studies in this manuscript confronted participants with two continuers. Future studies could increase the number of continuers. A different approach could focus on one continuer by either keeping a second continuer constant, or moving from paired comparisons to binary reidentification. Previous studies have explored variants of thought experiments compatible with these ideas. White [157] implemented one such scenario, focusing on the likelihood that a living person might be the reincarnation of a deceased person (see also [65]). For reincarnation judgments, distinctiveness was found to guide decisions. Similar to our sci-fi scenario, this setting might introduce specific assumptions about the process of reincarnation that could guide responses. For example, the importance of body similarity might be evaluated to be a lot lower than when responding to a scenario, in which a ship wreck survivor is returned from an island and matched to missing persons, and the importance of moral attributes to be higher. In contrast to the second scenario, the reincarnation scenario prevents the use of causal histories that are useful for person tracking [1].

It might also be the case that different practical concerns demand different criteria of identity. We investigated the parameters of identity in the context of re-identification and compensation. Other practical concerns, such as attributing blame, responsibility, or guilt, or allocating punishments and rewards, might trigger different responses, as the criteria of identity might shift or current properties of persons might become more relevant than historical properties and re-identification questions [10, 158].

To sum up, while we made progress to shed light on the decision processes used by participants, we have not yet established a complete process model, which should be the goal for future research [159, 160].

## Does it matter what matters for reidentification?

What are the implications of our studies for debates in cognitive sciences and other disciplines? Lay intuitions may be more prone to error than those of philosophers [100]—although experts' authority may also be questionable, as their philosophical intuitions are partly a function of their personality [161]. Our participants' endorsement of identity relations of an original with two non-identical persons violates the transitivity of identity. Yet this response pattern may simply show that our participants do not conceive of personal identity as strictly numerical, or that they have alternative conceptions of persons. Our results are in any case highly relevant for a descriptive analysis of people's understanding of identity and their theories of survival. Further, Nozick [14] would not attribute the variance in people's perspectives to errors or false everyday beliefs [162], but rather to the variance in closeness metrics legitimately deemed appropriate by different persons. Our findings are similarly relevant for cases in which scientists, philosophers, or marketers try to appeal directly to lay intuitions or common sense.

Philosophers and psychologists differ in their conceptualizations of intuitions [163], yet in our complex scenarios participants could not arrive at their answers without careful assessment. When appealing to the common sense of people both in theorizing and in legitimizing operationalizations, a researcher "should respect what ordinary people in fact say when asked —unless they are somehow led astray" (in [115], p. 216). Study 2 eliminated one way in which participants may have been led astray, by moving from a continuous to an all-or-nothing decision. Any appeal to general intuition should take into account both our main results and the interindividual variability demonstrated in both studies. Any attempt to measure the

conceptualization of self or personal identity may be informed by both our positive and our negative findings.

Psychological research has analyzed psychopathological conditions that entail potential breaks in personal continuity and identity [164]. For example, patients with the Fregoli delusion are convinced that different people are in fact a single person appearing in a variety of disguises. Here, the recognition of the outward appearance is separated from the identification of the person. This disorder goes beyond prosopagnosia, or face blindness, where the perception of faces does not allow for identification of persons (see also [165]). Patients with Capgras delusion believe that a specific person, often a loved one, has been replaced by a duplicate who is indistinguishable from the original person. In cases of mirror misidentification, patients fail to recognize their own reflection and infer that the person in the mirror must be someone else [166]. These observations from clinical psychology indicate that the neural mechanisms of identification cannot be reduced to acts of perceptual recognition, and hint at the requisite capacities necessary for personal reidentification; furthermore, understanding ordinary reidentification processes might help to understand and locate their disruption.

As the introductory example shows, the importance and reach of identity questions are not limited to specialized academic discourse, even if not every instance is as dramatic as the hanging of Arnaud du Tilh. A survey of current debates outside philosophy referencing the personal identity literature creates the impression that many of Parfit's [6] suggestions, examples and ideas are still in the process of being (re)discovered. Mott [167] explored Parfit's suggestion that diminished personal connectedness might be a reason for statutes of limitations (p. 325), and provided evidence that desert of punishment and grounds for criticizing a person for past deeds are considered to diminish over time, which is partially explained by the reduction in closeness. A second debate is centered on the question of the validity of living wills after substantive changes to a person's cognitive capabilities. For example, can a competent person impose values and interests on the future incompetent person or should the strength of advance directives grow weaker with the loss of closeness [21, 81, 168]? The incompetent person might, for example, derive unexpected pleasure and satisfaction under conditions the competent person did not foresee. Further, references to a future self might have a tremendous impact on ethical behavior [46], motivation, and goal-pursuit [169]; the sense of temporal persistence can motivate future-oriented self-regard and short-term sacrifices benefiting future outcomes [43, 44]. Without personal identity, it would be meaningless to make promises, grant ownership or the right to vote [79], or offer compensation [10, 12]; challenges to personal identity affect the institutions built upon it.

Disruptions of personal identity have been shown to severely impact people's lives. Chandler and colleagues [61] presented impressive evidence of the connection between the inability to give an account of one's identity and the risk of adolescent suicide, and how cultural continuity can moderate the elevation of suicide risk in vulnerable minority groups. This line of research raises the question or how one's perspective on personal identity is shaped by the social environment and connects the concept to mental health. A diminished sense of self and the self's stable existence is also deeply intertwined with borderline personality disorder [170]. Furthermore, challenges to personal identity can also emerge due to technological innovation. Notions of identity are fundamental in conceptualizing behavior in virtual environments [171] and have implications in law in connection with identity theft [172] or impersonation. Pascalev and colleagues [63] discussed how first suggestions for a medical head transplant procedure introduced questions of personal identity into neuroethics.

The gravity of these real-world examples goes well beyond that found in hypothetical thought experiments. The analysis of contrafactual scenarios has nevertheless paved the way for addressing real-world concerns and situations, whose connection to personal identity is

discovered through analogy, created through technology, or bestowed by social institutions. Understanding how the concept is perceived and applied and how experimental puzzles of seemingly little direct relevance are tackled and solved can ultimately inform practitioners and theoreticians facing recurrent and novel situations with serious consequences.

## Supporting information

**S1 Supporting Information. Supporting material for "Putting your money where your self is".** Study materials, supporting analyses, and qualitative coding scheme.
(PDF)

## Acknowledgments

We thank Susannah Goss and Deb Ain for their careful editing of this manuscript. We are grateful to Katarzyna Dudzikowska, Julia Eberhardt, Adam Feltz, Patricia Kanngiesser, Anastasia Kozyreva, and Christopher Olivola for discussions and comments at previous stages of the manuscript.

## Author Contributions

**Conceptualization:** Jan K. Woike, Bruce Hood.

**Data curation:** Jan K. Woike.

**Formal analysis:** Jan K. Woike.

**Investigation:** Jan K. Woike.

**Methodology:** Jan K. Woike.

**Software:** Jan K. Woike.

**Visualization:** Jan K. Woike.

**Writing – original draft:** Jan K. Woike.

**Writing – review & editing:** Jan K. Woike, Philip Collard, Bruce Hood.

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
