## [Decision Letter · Decision Letter 0]

30 Oct 2019

PONE-D-19-26910

Putting your Money Where your Self is: Connecting Dimensions of Closeness and Theories of Personal Identity

PLOS ONE

Dear Dr. Woike,

Thank you for submitting your manuscript to PLOS ONE. After careful consideration, we feel that it has merit but does not fully meet PLOS ONE’s publication criteria as it currently stands. Therefore, we invite you to submit a revised version of the manuscript that addresses the points raised during the review process.

Please find below the reviewers' comments.

We would appreciate receiving your revised manuscript by Dec 14 2019 11:59PM. To enhance the reproducibility of your results, we recommend that if applicable you deposit your laboratory protocols in protocols.io, where a protocol can be assigned its own identifier (DOI) such that it can be cited independently in the future. For instructions see: http://journals.plos.org/plosone/s/submission-guidelines#loc-laboratory-protocols

We look forward to receiving your revised manuscript.

Kind regards,

Valerio Capraro

Academic Editor

PLOS ONE

Journal Requirements:

1. We note that you have indicated that data from this study are available upon request. PLOS only allows data to be available upon request if there are legal or ethical restrictions on sharing data publicly. For more information on unacceptable data access restrictions, please see http://journals.plos.org/plosone/s/data-availability#loc-unacceptable-data-access-restrictions.

Additional Editor Comments (if provided):

I have now collected four reviews from four experts in the field. The reviewers like the paper but suggest several improvements before publication. Therefore, I would like to invite you to revise your work following the reviewers' comments. I am looking forward for the revision.

Reviewers' comments:

Reviewer's Responses to Questions

**Comments to the Author**

1. Is the manuscript technically sound, and do the data support the conclusions?

Reviewer #1: Partly

Reviewer #2: Yes

Reviewer #3: Yes

Reviewer #4: Partly

2. Has the statistical analysis been performed appropriately and rigorously? 

Reviewer #1: Yes

Reviewer #2: Yes

Reviewer #3: Yes

Reviewer #4: Yes

3. Have the authors made all data underlying the findings in their manuscript fully available?

Reviewer #1: Yes

Reviewer #2: No

Reviewer #3: No

Reviewer #4: Yes

4. Is the manuscript presented in an intelligible fashion and written in standard English?

Reviewer #1: Yes

Reviewer #2: Yes

Reviewer #3: Yes

Reviewer #4: Yes

5. Review Comments to the Author

Reviewer #1: This manuscript presents two experiments (with American sample from MTurk) that provide a welcome addition to the psychological literature on folk judgments of personal identity. It is also an important contribution to the broader theoretical discussion that is relevant to philosophers as well. Overall, I’m quite sympathetic towards the main goals of the studies and see some value in the methodological approach that has been taken (though with some issues), in particular, an attempt to apply the so-called fission cases to explore the patterns of folk personal identity judgments. However, the paper could use some conceptual polishing and a little more precision in the framing, but by and large, this study merits eventual publication.

1. General remarks.

A more general remark would be that the paper is rather dense and packed with a lot of (and less relevant) discussions that could be side-passed. The main claims could be presented in a more condensed form and more tied to the evidence produced. I understand that there are many strands of research and theorizing in this area, but not all of it is equally relevant to the claims of the current paper. The paper is massive and some tightening up would be desirable (especially introduction and general discussion sections).

Following on this general remark, there is one interesting feature to this paper. I am not sure what to make of it. That is, the authors throughout all the paper (and especially in the general discussions section) devote a considerable space to discuss possible objections and possible responses to those objections. For instance, in the general discussion section there is a whole rather long sub-section (“The fission scenario as scientific fiction: Benefits and validity concerns”) devoted to discuss the validity of the scenarios. The fact that the authors themselves took great pains to identify possible objections to their methodology is a great and welcome endeavor. But it does also show that there are some inherent problems in the employed methodology and, I must say, not all attempts to respond to objections worked very well. I guess a more honest strategy would be simply to acknowledge the problems and vow (so to say) to deal with them in the future studies. Actually, while reading the paper similar objections where popping up in my mind, and particularly in respect to methodology. Bellow I’ll raise some more concrete questions and concerns that could be addressed in the current paper or in the future studies.

2. Closest continuer theory and identity?

It is not clear how the proposed framework is different from (or similar to) Rips at el. (2006) proposal that also builds on Nizick’s theory. Some clarification would be useful. However, it should be mentioned that Rips at el. emphasized causality in their approach: “the continuer of the original object must be a causal outgrowth of that original” (Rips, Blok, & Newman 2006: 7). That is, perceived causal continuity is deemed to be central in producing and maintaining and an individual object over time and transformations. Correct me if I am wrong, but causality was not mentioned in the current formulation of the Closest continuer theory. Closeness metric, I assume, here is used to flesh out the theory. At any rate, if perceived causal continuity is deemed to be central for identity judgments, then this is a very general theory of rather thin individual identity (and not of rather thick personal identity). In other words, if my characterization is correct, what the proposed version of Closest continuer theory adds to the discussion about personal identity per se?

3. What “person” consists of?

Another questions is even more pressing and has some consequences for the methodology used. The authors differentiated five dimensions of the person and assumed that these five categories are the way how the folk conceptualize personhood. With all fairness, the authors did provide justifications for the choice of these dimensions, but all of it relies on psychological research and theories. What is more important in this context, I would argue, is how folk themselves conceptualize the personhood. So my question is, did the authors try to do pre-testing or surveys in order to extract the most salient dimensions from the people?

On the other hand, I am not sure whether it is warranted to have two psychology-related categories as separate dimensions: personality and psychology / memory and knowledge. This is clearly one category, but with some sub-categories. Besides, results indicate that both subcategories are the most significant dimensions in allocating insurance money. Also, in judgments of importance of these dimensions, both sub-categories correlate pretty well (eg., Table S5). Moreover, the choice not to include moral attributes is likewise not well justified (like Strohminger and Nichols did)

They write: “we did not differentiate between personality and psychology on the one hand and moral values on the other. In separate evaluations, our participants ranked memories and psychology as more important for identity than moral values.” (p. 32).

This post hoc justification doesn’t really help. At the outset, in determining dimensions of the personhood, it doesn’t matter how participants ranked an importance of psychology and moral values for identity. First, what matters is whether participants see it as a separate category that makes up the person. Besides, it looks quite strange when the authors distinguish possessions as a separate dimension (that, apparently, had no effect whatsoever), but excluded moral attributes.

Finally, it is not clear why for the category of “social relations” there was only “friends”. It is really good that the study had this dimension, but the scope of social relations has been rather limited. This doesn’t capture the whole complexity of social relations. Still, with all fairness, the authors did address this issues in Study 2, but again the choice of the items was not clear and transparent. Why this particular list of attributes?

4. Experimental scenarios.

Authors spent some time in justifying their choice of sci-fi scenarios. However, I share similar worries with critics who view such scenarios, either in philosophical thought experiments or in psychological experiments, with suspicion.

I must say that the sci-fi type of scenario is rather demanding on our imaginative capacities, there are way too many counterfactual elements that a participant has to keep in mind while “consulting” his/her intuitions on identity. Specifically, scenario starts with “hyperspace”: “You enter hyperspace to travel large distances and leave it at your new destination.”Then you are asked to imagine: “For a brief moment, the present universe overlaps with a parallel universe.” Finally, last section introduces:“Your travel agency contacts you while you are still in hyperspace and informs you that due to the overlap it has been calculated that, unfortunately, not one but two people will leave the hyperspace at your target destination: person A and person B, while you will no longer exist in your present state.”

I know people who don’t like sci-fi movies and admit that don’t understand them. So careful, unambiguous, construction of scenarios that is valid across different groups of people would be important. Did the authors make some comprehension pre-test (or some sort of cognitive interviewing)? If not, this is something for future research to consider.

Notwithstanding comprehensibility issues, I worry that the description of the scenario that started from 1st person perspective and ended with 3rd person descriptions of A and B persons might bias participants in some way. That is, it could have been the case that at least some participants looked at A and B persons as different from oneself since they were described from 3rd person perspective (this could also partially explain the predominant equal split, as if it was a split of money between, say, your kids, but not you in two different incarnations). Of course, it’s just a hunch, some other variants of the scenario need to be tested in order to rule out this possibility.

At any rate, authors claim that: “intuitions derived from cognitive processes during these unreal and outlandish examples may not be necessarily meaningful when measured against the constraints of reality but rather cast light on how we use and reason with the concept of identity.” (32). It might be the case. But a worry remains: does it really reveal any systematic and deeply-held intuitions/beliefs about personal identity or only post-hoc responses to rather weird scenarios. I might be wrong, but again the results indicated fair split of the money between A and B persons, which again points to some problems with scenarios. So, pre-testing is crucial here.

On the same note, there are some potential problems with post-questionnaire items:

1.1.5. Q9–Q13 — Importance for identity. How important are the following aspects for you when it comes to determining identity between two people?

This is an ambiguous question. One can read it as a question about qualitative identity between two numerically different people. As a result, some participants might have responded to this reading of the question.

1.1.6. Q14–Q18—Importance for survival. How important are the following aspects

for you when it comes to determining the survival of a person?

This is an ambiguous question as well. The notion of “survival” here is used in a more philosophical sense, but this is also an everyday term that has other semantic connotations. For instance, on could read it as a question about “what helps someone to stay alive” or “what helps someone to endure”. Thus, it is important to disambiguate between different readings. One can not assume that non-philosophers, ordinary folks, share similar understanding of the philosophical technical terms.

These potential problems in the methodology makes it hard to interpret results. However, I still belief that it is possible to extract some interesting and valuable information from the results as they are. While keeping the aforementioned concerns in mind (and presumably for future research), current results are consistent with previous work indicating that psychology is the most salient criterion in determining personal continuity, even in the case of fission. This is something akin to what Parfit advocated for — psychological connectedness. After some additional theoretical pruning, conceptual polishing and honest acknowledgment of some methodological issues (vowing to address them in the future), a revised version of the paper can be accepted for publication.

Reviewer #2: This paper investigates people’s beliefs about continuity of a person, empirically assessing the adequacy of Nozick’s closest continuer model and testing which bases seem to most inform continuity judgments.

I’ll be fairly brief because I think this paper is quite clear-cut. My understanding of the primary criteria for PLOS ONE is internal validity – are the conclusions well-supported by the data. I think the studies are well-designed, and the analysis is comprehensive and the conclusions are carefully drawn and accurately capture the results.

I’ll also add that I think the research question is of broad inter-disciplinary interest, and the paper does an excellent job of surveying that broad and inter-disciplinary literature, and the questions being asked have been debated but not answered in the prior literature. So, I think the contribution is quite clear.

There are some limitations of the paper. I think the primary limitation is that the paper uses highly unrealistic “science fiction” scenarios. I agree with the authors’ reasoning for doing this, and the paper includes an entire section at the end with a fair discussion of the benefits and limitations of this approach.

I was also a bit concerned about the characterization of the monetary allocation task as measuring only (or primarily) beliefs about continuity of identity. The discussion on p. 8-9 emphasizes the reasons why this measure should be responsive to beliefs about identity, but underplays a bit the possibility that other factors might influence the allocation decision. Given that dictator games typically reveal substantial sharing with distinct other people without a direct incentive to do so (e.g., as summarized in Engel 2011, which is cited), clearly other motivates can impact allocation decisions as well. The GD is more cautious on this point, and the other measures which do not have this issue reveal similar results, so I think this is a minor concern, but perhaps the initial discussion could be a bit clearer on this point.

Overall I think this is an important and well-done paper that will advance the debate about how people think of identity continuity.

My greatest concern, however, is that the authors do not plan to share their data. This is contrary to PLOS-ONE policy. It is the editor’s decision, but in my view, this is a serious issue as it relates to publication in PLOS ONE. The data in this paper is not the kind of sensitive or proprietary data that justifies an exemption, and IRB consent terms are, at least in part, a choice made by the authors. Deidentified data for minimal risk studies is commonly made public these days, and a long history in psychology has shown that “available upon reasonable request” policies simply do not work to ensure the necessary transparency. My opinion would typically be that if properly deidentified data (i.e., excluding not only identifiers but also demographics, if need be) cannot be posted, then the authors should submit the work to one of the journals that does not yet require data sharing.

However, I think this is somewhat of an atypical case. I noted the exhaustive online appendix, which is extraordinarily transparent about the data summaries and analyses. If the data absolutely cannot be posted, then I would suggest the following. First, I would like to see more detailed documentation on this point, to ensure that the data ethically cannot be shared (as opposed to the authors simply preferring not to share). Second, the online appendix should be permanently hosted on a public repository such as OSF or on PLOS ONE if published (i.e., as opposed to dropbox, where it might lapse into unavailability). Third, the full data analysis code, a data dictionary and a statement detailing the procedure for requesting the data including the timeframe for response should also be posted. I would also like the authors’ assurance that the data has already been prepared for sharing upon request.

Reviewer #3: This study on reactions to hypothetical cases of fission is a novel and important contribution to the literature on lay concepts of personal identity and survival. It presents important new results on empirically severely underexplored but theoretically highly significant branching thought experiments (especially notably, the study demonstrates that many participants identified the original with both continuers of the fission accident simultaneously); introduces methodological innovations; and enriches the empirical debate with important theoretical perspectives (namely, those by James and Nozick). I am confident that it should be published.

I have only two optional suggestions.

First, another body of literature in which structurally similar cases were explored in order to shed light on folk thinking about personal identity can be found in cognitive science of religion. Most notably, I would like to draw attention to work by Claire White on reincarnation beliefs (the most relevant paper is White, C. (2015). Establishing personal identity in reincarnation: Minds and bodies reconsidered. Journal of Cognition and Culture, 15(3-4), 402-429.). The experimental task in these studies is to identify the „true reincarnate“ from a set of potential candidates that vary in their features. These studies suggest a potential alternative explanation of present results - that it is not the type of content (e.g. presence of autobiographic memories vs presence of body) that drives reidentification judgments but differences in distinctiveness of ascribed features (and the more distinctive are the preserved features, the less likely it is that their presence is just a pure coincidence, the more attractive becomes an explanation of occurrence of these distinctive features in terms of preserved identity). It seems that some of the results of the present study could potentially be explained in this way (see e.g. Fig 7, where generic features, such as nationality, profession or group membership, seem to gravitate toward the ‘less important’ end of scale). It would be great if the authors briefly engaged with this potential alternative explanation.

Second, the authors write on p. 31 that ‘participants did not seem to differentiate between criteria for identity and criteria of survival’. While this is true for explicit weighting of dimensions, a quick glance at clusters in figures S13 and S16 suggests that for quite sizeable proportion of participants ascriptions of sameness and ascriptions of survival diverged markedly. Notably, in clusters A’ and D’ (approx. 40% of participants in Study 1) and A2’ in Study 2, participants ascribed survival and denied sameness. The same trend, while not crossing the midpoint (or crossing it less markedly) can be seen in several other clusters and also in mean agreements. It would be great to have additional stress (and additional comments on) the relationship between identity and survival (not only on explicit weighting of criteria but also on ascriptions of identity and survival themselves, where people seem to often differentiate between the two).

Reviewer #4: The manuscript explores continuity of personal identity using Nozick’s closest continuer theory. The manuscript examines a series of questions related to the theory including: What metrics determine closeness? Do people’s intuitions about continuity of identity conform to transitivity? Do identity judgments follow monetary allocation decisions? Do people follow the logic of Nozick’s theory? Do people’s explicit importance ratings predict monetary allocation decisions? In two studies, using a fission scenario, the manuscript explores these questions and some of the major finding are: memories/knowledge and personality/psychology tend to be the most important closeness metrics, people’s responses do not strictly adhere to transitivity of identity, judgments of identity tend to follow monetary allocations, explicit importance ratings predict monetary allocation decisions.

The manuscript does a nice job of examining some specific instances of whether Nozick’s theory describes how people think about personal identity (e.g., examining the violation of transitivity and similarity between identity judgments and monetary allocations) and clarifying what metrics are used in closeness judgments. However, because of the number of questions the manuscript aims to explore and the large number of results, it is a bit hard to follow, particularly the results section. The large number of results can make it hard to differentiate what the main findings are from what more peripheral findings are. Further, I think more clarity about what the goals of the research are and precisely what questions/results are relevant to each goal would be useful.

Major Comments:

It seems that there are at least three goals: 1) to explore and further specify what parts of Nozick’s theory mean—i.e., what determines people’s judgements of closeness, 2) to show that Nozick’s theory is a framework is a useful framework for describing how people think about personality identity and, 3) to use Nozick’s framework to resolve existing issues in the personal identity literature. While I think the design of the experiments and results seems to address the first goal, the support for the second and third goals would benefit from further support/clarification.

Regarding the second goal (to show that the closest continuer theory is a useful framework for describing how people think about personal identity), it seems like there are a number of areas where the results are counter to what the theory would predict or the relevant results are not highlighted. For example, the violation of transitivity seems to be in direct contradiction to the theory. Further, a key aspect of this theory seems to be that there is a separation between the continuer judgment and the closeness judgment but it’s not entirely clear to me where the separation in these types of judgments is shown in the participants’ responses.

The suggestion that the open-ended responses suggest that people follow the logic of the closest continuer theory needs further discussion. It appears that the evidence for this is that people tend to justify their allocation decisions with some statement about giving more money to the person who is more of them or closest to them. However, this seems like it could be consistent with a view of personality identity as similarity. It seems reasonable that “closest to me” or “more of me” could mean that people see that person as more similar and that people weight psychology and memory more heavily in similarity judgments than the other metrics used in these studies. Further, I would have thought that “logic” of the closest continuer theory would be more about the decision tree (and the separation between continuer and closeness judgments) rather than about just closeness alone as an input into allocation decisions. Perhaps this is addressed by the cluster analysis in the SM but it’s not clear to me which of those clusters correspond to those expected in Nozick’s decision scheme. ( As a result, I’m not sure what the statement “most responses can be interpreted meaningfully through the lens of the closest continuer theory” (pg. 19) refers to.

In general, as it is a complex theory and there are a large number of results, I think it’d be very helpful to, upfront, more clearly lay out the parts of the theory that are being tested and what pattern of results would support the theory and what pattern would be contradictory to the theory. Of course, the manuscript does lay out some of the major hypotheses and research questions at the beginning, however, some prioritization (or, perhaps, statement about what is not the focus of the manuscript) would be useful as to help focus the reader on the main takeaways.

Regarding the third goal, it seems like the suggestion is that Nozick’s framework is useful in resolving issues in the personal identity literature because it separates closeness and continuity (identity), and avoids the issue of studying similarity rather than identity. As noted above, it’s not entirely clear to me where the separation between closeness and continuity is shown in the results. It seems likely that there is data to show this but it either isn’t explicitly stated or is buried in a lot of other results.

I think the definition of similarity that is being used in this manuscript could be more explicitly stated as the difference between similarity and closeness is not entirely clear. One difference between closeness and similarity seems to be that closeness allows for different dimensions to have different weights and for the weights to vary across individuals. However, some accounts of similarity would allow for such differences in weights across features and across individuals (e.g., Ahn et al., 2000 Cognitive Psychology). If similarity is intended to mean simple feature overlap, since the “features” used in the study are such large categories (that may include differing numbers of features), it’s a bit unclear to me that the experiments can differentiate closeness from a feature-overlap similarity.

Minor Comments:

It would be very helpful to restate what the research questions are when they are referred to in the results section by number—it’s very hard to remember what question each number refers to. The hypotheses are re-stated more often, however, the same comment applies when they are only referred to by number.

6. PLOS authors have the option to publish the peer review history of their article (what does this mean?). If published, this will include your full peer review and any attached files.

Reviewer #1: No

Reviewer #2: No

Reviewer #3: No

Reviewer #4: No

---

## [Author Response · Author response to Decision Letter 0]

18 Dec 2019

Please see the attached document with responses to reviewers.

---

## [Decision Letter · Decision Letter 1]

13 Jan 2020

Putting your Money Where your Self is: Connecting Dimensions of Closeness and Theories of Personal Identity

PONE-D-19-26910R1

Dear Dr. Woike,

We are pleased to inform you that your manuscript has been judged scientifically suitable for publication and will be formally accepted for publication once it complies with all outstanding technical requirements.

With kind regards,

Valerio Capraro

Academic Editor

PLOS ONE

Additional Editor Comments (optional):

Reviewers' comments:

Reviewer's Responses to Questions

**Comments to the Author**

1. If the authors have adequately addressed your comments raised in a previous round of review and you feel that this manuscript is now acceptable for publication, you may indicate that here to bypass the “Comments to the Author” section, enter your conflict of interest statement in the “Confidential to Editor” section, and submit your "Accept" recommendation.

Reviewer #4: All comments have been addressed

2. Is the manuscript technically sound, and do the data support the conclusions?

Reviewer #4: Yes

3. Has the statistical analysis been performed appropriately and rigorously? 

Reviewer #4: Yes

4. Have the authors made all data underlying the findings in their manuscript fully available?

Reviewer #4: Yes

5. Is the manuscript presented in an intelligible fashion and written in standard English?

Reviewer #4: Yes

6. Review Comments to the Author

Reviewer #4: The revised manuscript includes many useful clarifications of both the results and the goals of the studies. The author's comments in their response letter were also useful in clarifying the goals of the manuscript. The reference to rational choice theory in the explanation of the research goals was very helpful and could potential be useful to readers if put in the manuscript. Overall, I am happy with the revision

7. PLOS authors have the option to publish the peer review history of their article (what does this mean?). If published, this will include your full peer review and any attached files.

Reviewer #4: No

---

## [Editor Report · Acceptance letter]

4 Feb 2020

PONE-D-19-26910R1 

Putting your Money Where your Self is: Connecting Dimensions of Closeness and Theories of Personal Identity 

Dear Dr. Woike:

I am pleased to inform you that your manuscript has been deemed suitable for publication in PLOS ONE. Congratulations! Your manuscript is now with our production department. 

With kind regards,

on behalf of

Dr. Valerio Capraro 

Academic Editor

PLOS ONE